# De novo gene signature identification from single-cell RNA-seq with hierarchical Poisson factorization

Hanna Mendes Levitin[1], Jinzhou Yuan[1], Yim Ling Cheng[1], Francisco JR Ruiz[2,3], Erin C Bush[1,4], Jeffrey N Bruce[5], Peter Canoll[6] (iD), Antonio Iavarone[6,7,8], Anna Lasorella[6,7,9], David M Blei[2,10,*] (iD) & Peter A Sims[1,4,11,**] (iD)

## Abstract

Common approaches to gene signature discovery in single-cell RNA-sequencing (scRNA-seq) depend upon predefined structures like clusters or pseudo-temporal order, require prior normalization, or do not account for the sparsity of single-cell data. We present single-cell hierarchical Poisson factorization (scHPF), a Bayesian factorization method that adapts hierarchical Poisson factorization (Gopalan *et al*, 2015, *Proceedings of the 31st Conference on Uncertainty in Artificial Intelligence*, 326) for *de novo* discovery of both continuous and discrete expression patterns from scRNA-seq. scHPF does not require prior normalization and captures statistical properties of single-cell data better than other methods in benchmark datasets. Applied to scRNA-seq of the core and margin of a high-grade glioma, scHPF uncovers marked differences in the abundance of glioma subpopulations across tumor regions and regionally associated expression biases within glioma subpopulations. scHFP revealed an expression signature that was spatially biased toward the glioma-infiltrated margins and associated with inferior survival in glioblastoma.

**Keywords** dimensionality reduction; gene signature discovery; glioma; single-cell RNA-seq
**Subject Categories** Chromatin, Epigenetics, Genomics & Functional Genomics; Genome-Scale & Integrative Biology
**Mol Syst Biol. (2019) 15: e8557**

## Introduction

Recent advances in the scalability of single-cell RNA-sequencing (scRNA-seq) offer a new window into development, the cellular diversity of complex tissues, cellular response to stimuli, and human disease. Conventional methods for cell-type discovery find clusters of cells with similar expression profiles, followed by statistical analysis to identify subpopulation-specific markers (Darmanis *et al*, 2015; Levine *et al*, 2015; Zeisel *et al*, 2015; Shekhar *et al*, 2016). Studies of cell fate specification have benefitted from innovative methods for inferring pseudo-temporal orderings of cells, allowing identification of genes that vary along a trajectory (Trapnell *et al*, 2014; Haghverdi *et al*, 2015; Setty *et al*, 2016). By design, these approaches discover expression programs associated with either discrete subpopulations or ordered phenotypes like differentiation status. However, in addition to cell type and developmental maturity, a cell's transcriptional state may include physiological processes like metabolism, growth, stress, and cell cycle; widespread transcriptional alterations due to copy number variants; and other co-regulated genes not specific to a discrete subpopulation or temporal ordering. Such expression programs are of particular interest in diseased tissue, where the underlying population structure may be unknown and druggable targets might vary independently of cell type or maturity.

Matrix factorization is an appealing approach to decomposing the transcriptional programs that underlie cellular identity and state without a predefined structure across cells. In this class of models, both cells and genes are projected into the same lower-dimensional space, and gene expression from each cell is distributed across

1   Department of Systems Biology, Columbia University Irving Medical Center, New York, NY, USA
2   Department of Computer Science, Columbia University, New York, NY, USA
3   Department of Engineering, University of Cambridge, Cambridge, UK
4   Sulzberger Columbia Genome Center, Columbia University Irving Medical Center, New York, NY, USA
5   Department of Neurological Surgery, Columbia University Irving Medical Center, New York, NY, USA
6   Department of Pathology and Cell Biology, Columbia University Irving Medical Center, New York, NY, USA
7   Institute for Cancer Genetics, Columbia University Irving Medical Center, New York, NY, USA
8   Department of Neurology, Columbia University Irving Medical Center, New York, NY, USA
9   Department of Pediatrics, Columbia University Irving Medical Center, New York, NY, USA
10  Department of Statistics, Columbia University, New York, NY, USA
11  Department of Biochemistry and Molecular Biophysics, Columbia University Irving Medical Center, New York, NY, USA
    *Corresponding author. Tel: +1 646 457 2110; E-mail: david.blei@columbia.edu
    **Corresponding author. Tel: +1 212 305 2687; E-mail: pas2182@cumc.columbia.edu

latent factors that approximate a vector basis for its transcriptional profile. Genes' weights over the latent factors are discovered simultaneously and can be used to identify expression programs. For example, previous studies have defined gene expression programs from scRNA-seq data using principal component analysis (PCA) or non-negative matrix factorization (NMF; Islam *et al*, 2011; Patel *et al*, 2014; Tirosh *et al*, 2016a; Chung *et al*, 2017; Puram *et al*, 2017). However, a combination of biological regulation, stochastic gene expression, and incomplete experimental sampling leads to sparsity in scRNA-seq data. This creates challenges in downstream analysis. Conventional methods like PCA and NMF are sensitive to false-negative dropout events in which a transcript is experimentally undetected despite its presence in a cell (Pierson & Yau, 2015; Prabhakaran *et al*, 2016). Further, sparsity may vary across both cells and genes, complicating the normalization that most computational methods require (Prabhakaran *et al*, 2016; Vallejos *et al*, 2017).

Here, we describe single-cell hierarchical Poisson factorization (scHPF), a Bayesian factorization method that uses hierarchical Poisson factorization (Gopalan *et al*, 2015) to avoid prior normalization and explicitly model variable sparsity across both cells and genes. We compare scHPF to popular normalization and dimensionality reduction methods as well as algorithms explicitly designed for scRNA-seq. scHPF has better predictive performance than these methods, more closely captures expression variability in datasets generated by multiple experimental technologies, and has better computational performance than other methods designed for scRNA-seq. Finally, we apply scHPF to single-cell expression profiles obtained from the core and invasive edge of a high-grade glioma. scHPF identifies both expected and novel features of tumor cells at single-cell resolution and uncovers an expression signature associated with poor survival in glioblastoma.

## Results

### Single-cell hierarchical Poisson factorization

scHPF uses hierarchical Poisson factorization (Gopalan *et al*, 2015) for *de novo* identification of gene expression programs from genome-wide unique molecular counts. In scHPF, each cell or gene has a limited "budget" which it distributes across the latent factors. In cells, this budget is constrained by transcriptional output and experimental sampling. Symmetrically, a gene's budget reflects its sparsity due to overall expression level, sampling, and variable detection. The interaction of a given cell and gene's budgeted loadings over factors determines the number of molecules of the gene detected in the cell.

More formally, scHPF is a hierarchical Bayesian model of the generative process for an $N \times M$ count matrix, where $N$ is the number of cells and $M$ is the number of genes (Fig 1). scHPF assumes that each gene $g$ and cell $i$ is associated with an inverse-budget $\eta_g$ and $\xi_i$, respectively, that probabilistically determines its observed transcriptional output. Since both $\eta_g$ and $\xi_i$ are positive-valued, scHPF places Gamma distributions over those latent variables. We set $\eta_g$ and $\xi_i$'s hyperparameters empirically (Materials and Methods, Fig EV1).

For each factor $k$, gene and cell loadings over factors $\beta_{g,k}$ and $\theta_{i,k}$ are drawn from a second layer of Gamma distributions whose rate

parameters depend on the inverse budgets $\eta_g$ and $\xi_i$ for each gene and cell. Setting these distributions' shape parameters close to zero enforces sparse representations, which can aid downstream interpretability. Finally, scHPF posits that the observed expression of a gene in a given cell is drawn from a Poisson distribution whose rate is the inner product of the gene's and cell's weights over factors. Importantly, scHPF accommodates the over-dispersion commonly associated with RNA-seq (Anders & Huber, 2010) because a Gamma-Poisson mixture distribution results in a negative binomial distribution; therefore, scHPF implicitly contains a negative binomial distribution in its generative process. Previous work suggests that the Gamma-Poisson mixture distribution is an appropriate noise model for scRNA-seq data with unique molecular identifiers (UMIs; Ziegenhain *et al*, 2017; preprint: Wagner *et al*, 2018).

Given a gene expression matrix, scHPF approximates the posterior distribution over the inverse budgets and latent factors given the data using coordinate ascent variational inference (Jordan *et al*, 1999; Blei *et al*, 2017; Materials and Methods). After fitting the model's variational posterior, we define each gene and cell's score for a factor $k$ as the expected values of its factor loading $\beta_{g,k}$ or $\theta_{i,k}$ times its inverse-budget $\eta_g$ or $\xi_i$, respectively, which scales scHPF's inferred loadings by its inferred budgets. We select the number of factors based on the convergence of the negative log likelihood and representation of major cell types (Materials and Methods).

Importantly, scHPF identifies factors *de novo* from genome-wide expression measurements. In this work, datasets include all protein-coding genes observed in at least $\sim 0.1\%$ of cells, typically $> 10,000$ genes (Appendix Table S1). In contrast, some previously published dimensionality reduction methods for scRNA-seq depend on preselected subsets of $\sim 1,000$ highly variable genes (which likely represent subpopulation-specific markers; Risso *et al*, 2018) or exclude genes observed in $< 5$–$10\%$ of cells (Pierson & Yau, 2015).

### Benchmarking against alternative methods

We compared scHPF's predictive performance to that of PCA, NMF, and factor analysis (FA), as well as two methods developed

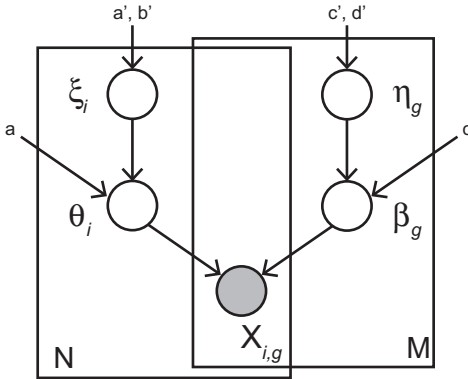

**Figure 1.    scHPF models the data matrix $X_{i,g}$ using a set of per-cell latent factors $\theta_i$ and per-gene latent factors $\beta_g$.**

scHPF places hierarchical priors over the latent factors through the latent variables $\xi_i$ and $\eta_g$, which probabilistically determine the observed transcriptional output for the cell or gene.

specifically for scRNA-seq: zero-inflated factor analysis (ZIFA; Pierson & Yau, 2015) and zero-inflated negative binomial-based wanted variation extraction (ZINB-WaVE; Risso *et al*, 2018). These methods have been used for expression program discovery without a predefined structure across cells (Patel *et al*, 2014; Pierson & Yau, 2015; Tirosh *et al*, 2016a; Chung *et al*, 2017; Puram *et al*, 2017). We assessed each method across three datasets in different biological systems and obtained with different experimental platforms (Appendix Table S1). The peripheral blood mononuclear cell (PBMC) data from 10× Genomics are a mixture of discrete cell types (Zheng *et al*, 2017a; Data ref. Zheng *et al*, 2017b), whereas the Matcovitch *et al*'s microglial dataset samples from multiple time-points along a developmental process (Matcovitch-Natan *et al*, 2016a; Data ref. Matcovitch-Natan *et al*, 2016b). Additionally, we profiled 9,924 cells from a patient-derived glioma neurosphere line (TS543), in which physiological processes like cell cycle, rather than discrete cell types or differentiation status, drive expression variability. The datasets originate from different biological systems and experimental technologies including droplet-based 10× Chromium (Zheng *et al*, 2017a), MARS-seq (Jaitin *et al*, 2014), and an automated microwell platform (Yuan & Sims, 2016).

For each dataset, we tested conventional methods with three different normalizations: log-transformed molecular counts, counts per median (rate normalization), and log-transformed counts per median (log-rate normalization). ZIFA was only evaluated using log-transformed normalizations as recommended by its authors, and ZINB-WaVE was applied directly to molecular counts. We did not apply ZINB-WaVE to the nearly 10,000-cell TS543 dataset due to the method's prohibitive computational cost (Table 1). With only one exception, scHPF had the best predictive performance on held-out test data across all datasets and normalizations (Fig 2A). scHPF's superior performance was robust across a range of values for $K$, the number of factors (Fig EV2). Notably, while ZINB-WaVE had better predictive performance than scHPF on PBMCs, it had the highest mean-squared error of any method on the Matcovitch *et al*'s dataset.

Because scHPF and ZINB-WaVE performed comparably in terms of predictive performance on the PBMC dataset, we carefully examined their respective factorizations in terms of computational expense and biological interpretability. For a single initialization

**Table 1.** Runtime and peak memory consumption at different levels of parallelization for ZINB-WaVE and scHPF on the PBMC dataset with $K = 10$.

|  | Parallelization | Runtime | Peak Memory |
|---|---|---|---|
| ZINB-WaVE | 1 vCPU | 442.58 min (7.38 h) | 17.8 Gb |
| ZINB-WaVE | 2 vCPU | 237.76 min (3.96 h) | 31.5 Gb |
| scHPF | 2 vCPU | 10.66 min | 1.6 Gb |
| scHPF | 8 vCPU | 6.13 min | 1.6 Gb |
| scHPF | 16 vCPU | 2.47 min | 1.7 Gb |
| scHPF (10 trials) | 8 vCPU | 52.90 min | 1.6 Gb |
| scHPF (10 trials) | 16 vCPU | 36.45 min | 1.6 Gb |

ZINB-WaVE's high memory consumption precluded running it with more than two threads on this dataset.

with $K = 10$, training ZINB-WaVE took 7.38 h and had a peak memory consumption of 17.8 Gb (Table 1). Using two threads reduced ZINB-WaVE's runtime to just under 4 h, but nearly doubled its memory consumption to 31.5 Gb. In contrast, our scHPF implementation took 2.5–10.7 min, depending on the number of threads available, and ~ 1.6 Gb of memory (Table 1). scHPF's superior performance is in part due to optimized compilation and automatic parallelization with the Python Numba library (Lam *et al*, 2015). In addition, unlike ZINB-WaVE, scHPF only needs to consider non-zero matrix entries during training (Materials and Methods), which imparts a considerable theoretical advantage over methods that must iterate through (and in some cases store) every matrix entry.

We compared the interpretability of scHPF and ZINB-WaVE's low-dimensional representations of cells in the PBMC data. Clustering using a conventional pipeline identified major PBMC types including monocytes, dendritic cells, T cells, and B cells (Materials and Methods, Fig EV3A). scHPF factors were in excellent agreement with clustering results (Fig EV3B and C). Each major cell type had an associated dominant factor, and there were relationships between factors associated with related cell types. In contrast, factors obtained using ZINB-WaVE did not exhibit the same close relationship to basic cell types in the data. While there were dominant factors for monocytes and B cells, smaller clusters did not relate to ZINB-WaVE factors in an interpretable way.

In bulk RNA-seq, modeling over-dispersed gene expression data has proven essential to downstream analysis (Anders & Huber, 2010). In scRNA-seq, expression data are over-dispersed both across genes in individual cells and for individual genes across cells. We evaluated how well different factorization methods captured single-cell expression variability using a posterior predictive check (PPC). PPCs provide insight into a generative model's goodness of fit by comparing the observed dataset to simulated data generated from the model. More formally, PPCs sample simulated replicate datasets $X_{rep}$ from a generative model's posterior predictive distribution and use a modeler-defined test statistic to evaluate discrepancies between $X_{rep}$ and the true data, $X_{obs}$ (Gelman *et al*, 2013). For each dataset, normalization, and generative factorization method (scHPF, PCA, FA, and ZIFA), we sampled ten replicate expression vectors per cell. After converting samples from models on normalized data back to molecular counts (Materials and Methods), we computed the coefficient of variation (CV) for all genes in each cell and each gene across all cells. Finally, we averaged each cell and gene's CVs across the ten replicate simulations. In all three datasets, scHPF more closely matched the observed data's variability than other methods (Fig 2B, Appendix Fig S2). We noticed that many samples from PCA and FA had physically impossible negative values. When we corrected these values by clipping them to zero, PCA and FA's estimates of variability across cells collapsed toward zero (Fig 2C). This collapse suggests that PCA and FA's ability to model over-dispersion in scRNA-seq data depends on placing probability mass on negative gene expression levels.

## Application to spatially sampled scRNA-seq from high-grade glioma

As a demonstration, we applied scHPF to 6,109 single-cell expression profiles from the core and invasive edge of a high-grade glioma. High-grade gliomas (HGGs), the most common and lethal brain

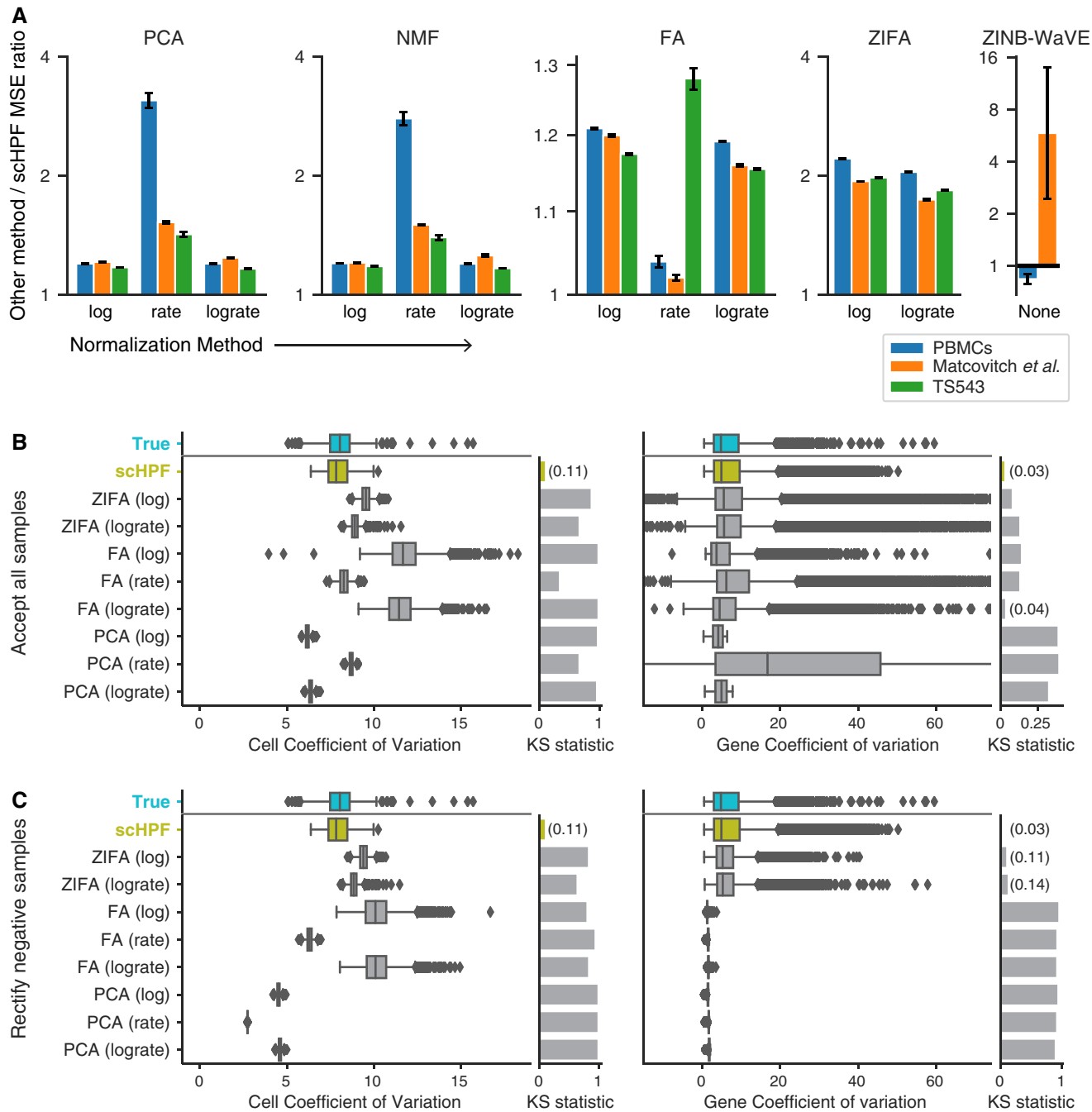

**Figure 2. scHPF captures statistical properties of scRNA-seq data better than alternative factorization methods.**

A  Ratio of mean-squared error (MSE) of different factorization methods on withheld test sets to scHPF's. scHPF's MSE was calculated after normalizing its predictions. Error bars show standard error of the mean across three train/validation/test splits; center values show the mean (Materials and Methods).

B  Posterior predictive checks of expression variability in PBMCs. Box plots show the coefficient of variation (CV) for gene expression within single cells across all genes (left) and for single genes across all cells (right) in both the true distribution (green) and posterior predictive simulations. X-axes limits are set to include all CVs from the true distribution and scHPF, and as many CVs from other methods as possible. Accompanying bar graphs show the maximum distances between the cumulative distributions of the true and simulated CV values (the Kolmogorov–Smirnov (KS) statistic, lower is better).

C  Same as (B), but clipping impossible negative posterior predictive samples to zero.

malignancies in adults (Ostrom *et al*, 2017), are highly heterogeneous tumors with complex microenvironments. In HGG, malignant cells invade the surrounding brain tissue, forming diffusely infiltrated margins that are impossible to fully remove surgically (Gill

*et al*, 2014). Although malignant cells in margins seed tumor recurrence and are the targets of post-operative therapy, most molecular characterization has focused on HGG cores. To investigate the transcriptional differences between cells in glioma's core and margins,

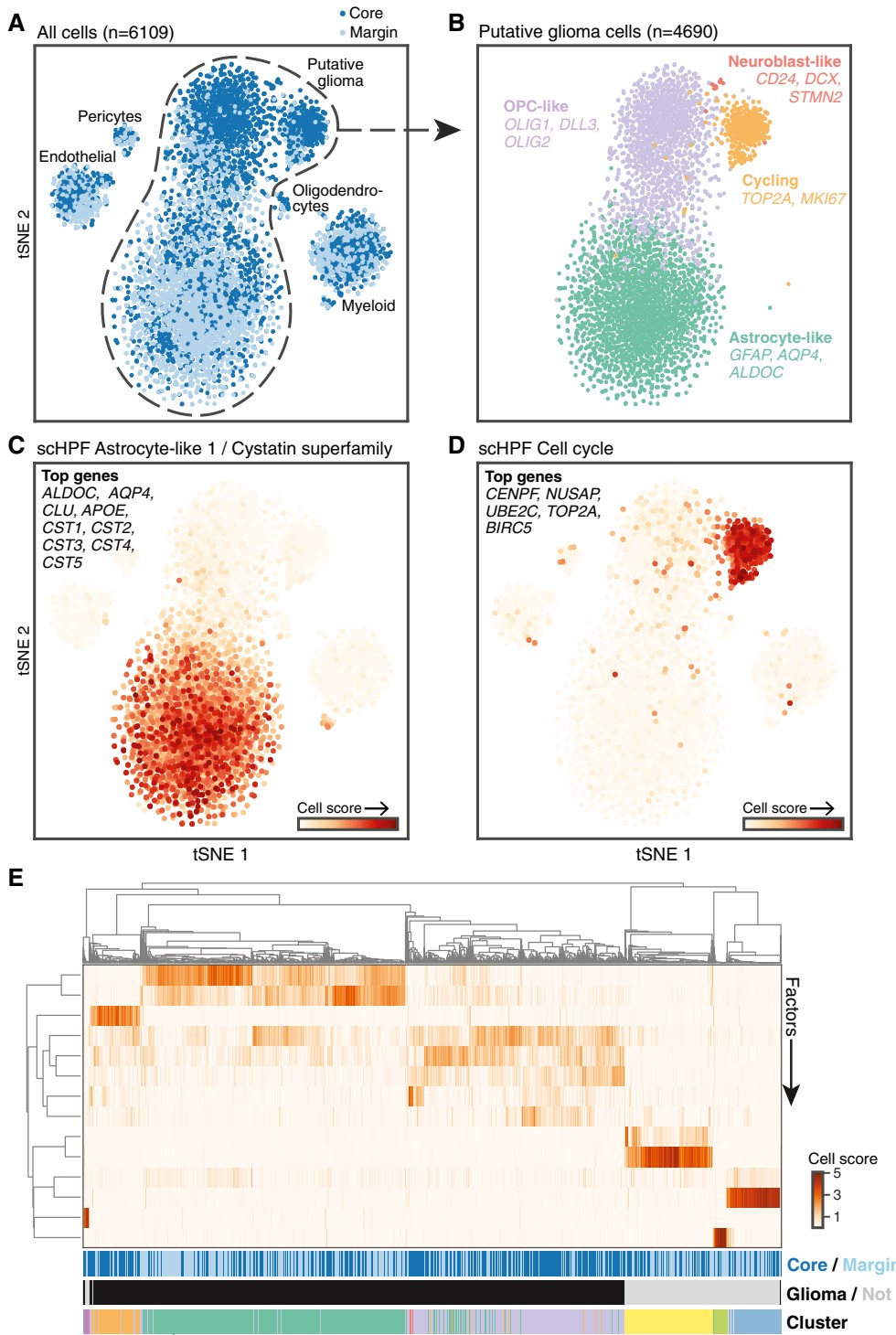

**Figure 3.   scHPF agrees with conventional analysis for regionally identified scRNA-seq of a high-grade glioma (HGG).**

A    t-Distributed stochastic neighbor embedding (tSNE; Maaten & Hinton, 2008) plot of cells from the core (navy) and margin (light blue) of an HGG reveals both malignant and non-malignant subpopulations (Materials and Methods). Labels were determined using malignancy score, clustering, and differential expression (Appendix Figs S2 and S3, Materials and Methods).

B    tSNE representation of putative glioma cells colored by cluster highlights astrocyte-like, OPC-like, neuroblast-like, and cycling subpopulations.

C    tSNE representation of all tumor cells colored by scHPF cell scores for one of two astrocyte-like factors. Nine out of the top 30 highest scoring genes are highlighted.

D    Same as (C), but for a cell cycle factor identified by scHPF. The five top-scoring genes in the factor are listed.

E    Main heatmap shows hierarchical clustering of cells' scores for each factor. Top colorbar indicates the cell's region: core (navy) or invasive edge (light blue). Second colorbar shows putative neoplastic status. Bottom colorbar indicates cluster.

we used an MRI-guided procurement technique (Gill *et al*, 2014) and scRNA-seq to profile 3,109 cells from an HGG core and 3,000 cells from its margin. While recent efforts are beginning to shed light on the differential expression between glioma's core and margins (Gill *et al*, 2014; Darmanis *et al*, 2017), few studies involve this kind of spatial sampling.

Glioma cells typically resemble glia at the level of gene expression, and our prior work characterizing HGGs with scRNA-seq revealed co-occurring malignant subpopulations resembling astrocytes, oligodendrocyte progenitors (OPCs), and neuroblasts (Yuan *et al*, 2018). Consistent with these findings, clustering and aneuploidy analysis (Materials and Methods, Appendix Figs S2 and S3) revealed malignant subpopulations that expressed markers of astrocytes, OPCs, neuroblasts, and dividing cells as well as non-malignant populations of myeloid cells, oligodendrocytes, endothelial cells, and pericytes (Fig 3A and B, Appendix Figs S2 and S3). In the spatially resolved samples, malignant subpopulations had dramatically different abundances across regions (Appendix Fig S3H). Astrocyte-like glioma cells were over twofold more abundant in the margin biopsy, while OPC-like and cycling populations were nearly threefold and fourfold better represented in the core biopsy. All seventeen neuroblast-like glioma cells localized to the tumor core.

Applied to the same dataset, scHPF identified at least one factor associated with every cell type, as well as physiological processes like translation, cell cycle, and stress response (Figs 3C and D, and EV4, Appendix Table S2). Cell's scHPF scores were largely uncorrelated with technical variables (Appendix Fig S4); however, two factors associated with physically larger cell types (dividing and endothelial) were modestly correlated with the number of molecules and genes per cell. Hierarchical clustering of cells' scores across factors recapitulated both Louvain clustering and malignant status (Fig 3E), and factors associated with malignant subpopulations had regional biases across glioma cells that were consistent with glioma subpopulations' differential abundance across regions (Appendix Fig S5A). In addition, we could use scHPF's factorization as a low-dimensional input to t-distributed stochastic neighbor embedding (tSNE; Maaten & Hinton, 2008) or uniform manifold approximation and projection (UMAP; McInnes *et al*, 2018) to produce visualizations that were consistent with conventional clustering (Fig EV5). Taken together, these results show that scHPF captures the major features identified by standard analyses of this dataset.

Some scHPF factors' scores varied within the subpopulations identified by clustering. For example, two myeloid-associated factors that ranked pro-inflammatory cytokines and S100-family genes highly (Fig EV4A), respectively, were correlated across all cells ($r = 0.66$, $P < 10^{-100}$) but anticorrelated within the myeloid cluster ($r = -0.59$, $P < 10^{-71}$). Together, they appeared to represent a continuum of immune activation (Fig 4A–C). This phenotypic gradient within an individual tumor is reminiscent of the variable myeloid states observed across different patients in previous studies of glioma (Tirosh *et al*, 2016b; Müller *et al*, 2017; Venteicher *et al*, 2017; Yuan *et al*, 2018).

While scHPF factors had regional biases that reflect overall compositional differences between the core and margin biopsies, glioma cells' scHPF factor scores also exhibited regional biases *within* the malignant subpopulations defined by clustering (Fig 4D–

F, Appendix Fig S5A). For example, OPC-like glioma cells in the tumor core had significantly higher scores for the neuroblast-like, OPC-like, and cell cycle factors than their counterparts in the margin (Bonferroni corrected $P < 10^{-84}$, $P < 10^{-6}$, and $P < 10^{-6}$, respectively, by the Mann–Whitney *U*-test), whereas OPC-like glioma cells in the margin had higher scores for the two astrocyte-like factors ($P < 10^{-49}$ and $P < 10^{-69}$ for astrocyte-like factors 2 and 1, respectively). These differences were driven by the highest scoring genes in each factor (Appendix Fig S5B), and astrocyte-like glioma cells followed a similar pattern. An alternative method of determining cellular subpopulations, where cells were assigned to the subpopulation with which their highest scoring factor was associated, also preserved the regional biases (Appendix Fig S5C). This analysis suggests that, in this case, cells in the same malignant subpopulations but different tumor regions may have subtly different lineage resemblances.

As cells from the HGG margin remain after surgery and seed aggressive recurrent tumors, we investigated whether regionally biased transcriptional signatures derived from scHPF factors were associated with survival in The Cancer Genome Atlas (TCGA) (Verhaak *et al*, 2010). Restricting the analysis to glioblastoma (GBM), we identified patients enriched and depleted for the top genes in each factor (Materials and Methods, Appendix Fig S6 for analysis of sensitivity to effect size thresholds). Survival analysis revealed significantly shorter overall survival ($\sim$ 1 year median difference) for patients enriched for a margin-biased scHPF astrocyte-like signature (Fig 4G and H), which included astrocytic markers *ALDOC*, *CLU*, and *SPARCL1* (Bachoo *et al*, 2004; Zhang *et al*, 2014, 2016), as well as cystatin super-family members *CST1* though *CST5* (Figs 3C and EV4A). Cystatin C (*CST3*) is highly expressed in mature human astrocytes (Bachoo *et al*, 2004; Zhang *et al*, 2016) and is induced in Alzheimer's disease and epilepsy (Steinhoff *et al*, 2001; Pirttilä *et al*, 2005; Gauthier *et al*, 2011), raising the possibility that astrocyte-like glioma cells may be responding to the same cues or stresses that reactive astrocytes encounter in these disorders. Although it is difficult to determine which cells are responsible for an expression signature in bulk RNA-seq data, top scHPF astrocyte-like factor 1 genes were better correlated with molecular markers of tumor cells than other cells in the tumor microenvironment (Appendix Fig S7), suggesting that glioma cells express those genes.

## Discussion

Conventional approaches to analyzing scRNA-seq data use predefined structures like clusters or pseudo-temporal orderings to identify discrete transcriptional programs associated with particular subpopulations and pseudo-temporally coupled gene signatures. However, gene expression programs may vary independently of these structures across complex populations. scHPF complements conventional approaches, allowing for *de novo* identification of transcriptional programs directly from a matrix of molecular counts in a single pass. By explicitly modeling variable sparsity in scRNA-seq data and avoiding prior normalization, scHPF achieves better predictive performance than other *de novo* matrix factorization methods while also better capturing scRNA-seq data's characteristic variability.

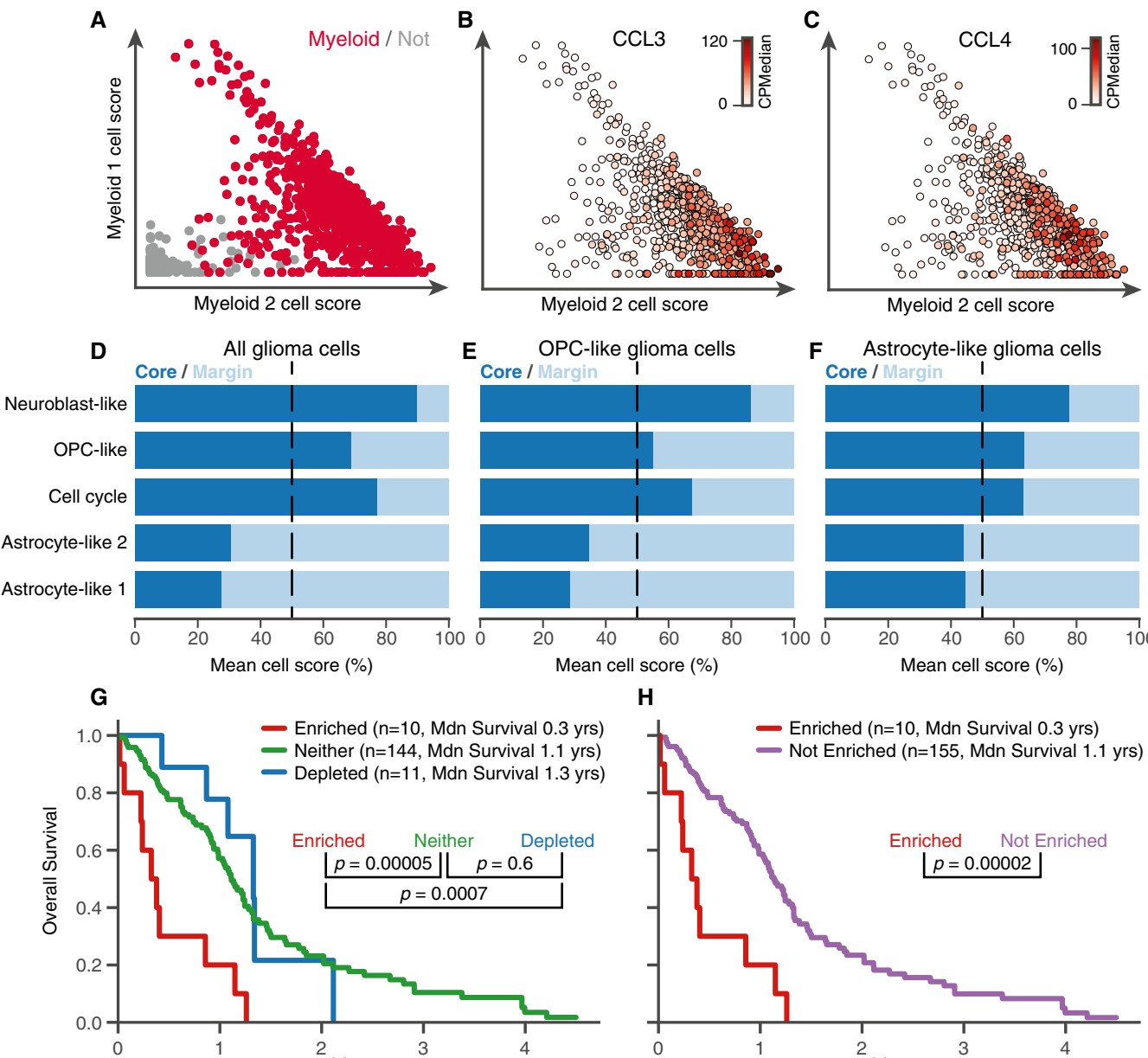

**Figure 4.  scHPF identifies finely resolved and novel, regionally associated features of HGG.**

A       Scores for myeloid factor 1 (*y*-axis) versus myeloid factor 2 (*x*-axis) for cells in the myeloid Louvain cluster (crimson) and all other cells (gray).

B, C    Expression of pro-inflammatory cytokines CCL3 (B) and CCL4 (C) for cells in the myeloid subpopulation shows a gradient of activation.

D–F     Factor score bias between the core (navy) and margin (light blue) in all glioma cells (D), OPC-like glioma cells (E), and astrocyte-like glioma cells (F). Mean cells scores in each region are scaled to sum to 100. Biases are driven by the top genes in each factor (Appendix Fig S6D–F).

G, H    Kaplan–Meir curves show overall survival differences in TCGA for donors enriched (red), not enriched (purple), depleted (blue), or neither enriched nor depleted (green) for the 25 top-scoring genes in astrocyte-like factor 1 (Materials and Methods, Appendix Fig S6 for different effect size cutoffs). *P*-values were computed using a log-rank test and Mdn survival indicates median overall survival.

In scRNA-seq of biopsies from the core and margin of a high-grade glioma, scHPF recapitulated and expanded upon molecular features identified by standard analyses, including expression signatures associated with all of the major subpopulations and cell types identified by clustering. Importantly, some lineage-associated factors identified by scHPF varied within or across clustering-defined populations, revealing features that

were not apparent from cluster-based analysis alone. Clustering analysis showed that astrocyte-like glioma cells were more numerous in the tumor margin while OPC-like, neuroblast-like, and cycling glioma cells were more abundant in the tumor core. scHPF not only recapitulated this finding, but also illuminated regional differences in lineage resemblance within glioma subpopulations. In particular, both OPC-like and astrocyte-like

glioma cells in the tumor core had a slightly more neuroblast-like phenotype than their more astrocyte-like counterparts in the margin. Finally, we discovered a margin-biased gene signature enriched among astrocyte-like glioma cells that is highly deleterious to survival in GBM.

Massively parallel scRNA-seq of complex tissues in normal, developmental, and disease contexts has challenged our notion of "cell type" (Wagner *et al*, 2016), particularly as highly scalable methods provide ever-increasing resolution. Further, gene expression programs essential to tissue function may be highly cell type-specific or might vary continuously within or across multiple cell types. Conventional graph- and clustering-based methods provide invaluable insight into the structure of complex cellular populations, and much can be learned from projecting single-cell expression profiles onto these structures. scHFP effectively models the nuanced features of scRNA-seq data while identifying highly variable gene signatures, unconstrained by predefined structures such as clusters or trajectories. We anticipate that scHFP will be a complementary tool for dissecting the transcriptional underpinnings of cellular identity and state.

# Materials and Methods

### Reagents and Tools table

| Reagent/Resource | Reference or Source | Identifier or Catalog number |
|---|---|---|
| **Experimental models** | | |
| *TS543* | Memorial Sloan Kettering | |
| Radiographically guided biopsies of high-grade glioma | Columbia University Medical Center | |
| **Oligonucleotides and sequence-based reagents** | | |
| Template switch oligo (AAGCAGTGGTATCAACGCAGAGTGAATrGrGrG) | IDT | |
| SMRT PCR primer (AAGCAGTGGTATCAACGCAGAGT) | IDT | |
| Custom P5 nextera pcr primer (AATGATACGGCGACCACCGAGATCTACACGCCTGTCC GCGGAAGCAGTGGTATCAACGCAGAGT*A*C) | IDT | |
| Custom read 1 sequencing primer (GCCTGTC CGCGGAAGCAGTGGTATCAACGCAGAGTAC) | IDT | |
| **Chemicals, enzymes and other reagents** | | |
| 10× Tris Buffered Saline (TBS) solution | Sigma | T5912 |
| Exo I | NEB | M0293S |
| SuperaseIN | ThermoFisher | AM2696 |
| Maxima H- Reverse Transcriptase | ThermoFisher | EP0752 |
| KAPA HiFi HotStart ReadyMix | Kapabiosystems | KK2602 |
| Calcein AM cell-permeant dye | ThermoFisher | C3100MP |
| 2-Mercaptoethanol | ThermoFisher | BP176-100 |
| Fluorinert Oil Fluorinert® FC-770 | Sigma | F3556-25ML |
| Buffer TCL | Qiagen | 1031576 |
| Heparin | Stem Cell Technologies | 7980 |
| NeuroCult NS-A Proliferation Supplement | Stem Cell Technologies | 05751 |
| NeuroCult NS-A Basal Medium | Stem Cell Technologies | 05750 |
| Human recombinant epidermal growth factor (EGF) | Stem Cell Technologies | 78006.1 |
| Human recombinant basic fibroblast growth factor (bFGF) | Stem Cell Technologies | 78003 |
| Dow Corning Sylgard 184 Silicone | ESSEX BROWNELL | DC-184-1.1 |
| **Software** | | |
| samtools v1.3 | https://github.com/samtools/samtools/tree/1.3 | |
| bwa-mem v0.7.12 | https://github.com/lh3/bwa/tree/0.7.12 | |
| bedtools v2.17.0-1 | https://launchpad.net/ubuntu/+source/bedtools/2.17.0-1 | |
| STAR v2.5.0 | https://github.com/alexdobin/STAR/tree/2.5.0b | |

**Reagents and Tools table**   (continued)

| Reagent/Resource | Reference or Source | Identifier or Catalog number |
|---|---|---|
| **Other** | | |
| Nextera XT DNA Library Preparation Kit (24 samples) | Illumina | FC-131-1024 |
| Nextera XT Index Kit (96 Indexes, 384 Samples) | Illumina | FC-131-1002 |
| Dropseq beads (barcoded mRNA capture beads) | ChemGenes | MACOSKO-2011-10 |
| Ampure beads | Beckman Coulter | A63880 |
| Qubit® dsDNA HS Assay Kit | Life Technologies | Q32851 |
| High Sensitivity DNA chips kit | Agilent Technologies | 5067-4626 |
| ZR-Duet™ DNA/RNA MiniPrep | Zymo Research | D7001 |
| NextSeq 500/550 High Output v2 kit (150 cycles) | Illumina | FC-404-2002 |
| NextSeq 500/550 High Output v2 kit (75 cycles) | Illumina | FC-404-2005 |
| PhiX Sequencing Control | Illumina | FC-110-3001 |

## Methods and Protocols

### Single-cell hierarchical Poisson factorization

The generative process for single-cell hierarchical Poisson factorization, illustrated in Fig 1, is as follows:

1  For each cell $i$:
   a  Sample capacity $\xi_i \sim \text{Gamma}(a', b')$
   b  For each factor $k$:
      i  Sample weight $\theta_{i,k} \sim \text{Gamma}(a, \xi_i)$
2  For each gene $g$:
   a  Sample capacity $\eta_g \sim \text{Gamma}(c', d')$
   b  For each factor $k$:
      i  Sample weight $\beta_{g,k} \sim \text{Gamma}(c, \eta_g)$
3  For each cell $i$ and gene $g$, sample observed expression level $x_{i,g} \sim \text{Poisson}(\theta_i \beta_g^T)$,

where $x$ is a discrete scRNA-seq expression matrix.

For *de novo* gene signature identification, we define each cell $c$'s score for each factor $k$ as

$$\text{cell\_score}_{i,k} = E[\xi_i|\boldsymbol{x}] * E[\theta_{i,k}|\boldsymbol{x}]$$

and each gene $g$'s score for each factor $k$ as

$$\text{gene\_score}_{i,k} = E[\eta_g|\boldsymbol{x}] * E[\beta_{g,k}|\boldsymbol{x}].$$

This adjusts factor loadings for the learned transcriptional output of their corresponding cell or gene. Finally, we rank the genes in each factor by their scores to identify *de novo* patterns of coordinated gene expression (e.g. Fig EV4A). Cell's scores, for example, those plotted Figs 3C and D, and EV4B–D, indicate a cell's association with the factor.

### Inference

We use coordinate ascent variational inference to approximate $p(\xi, \boldsymbol{\theta}, \boldsymbol{\eta}, \boldsymbol{\beta}|\boldsymbol{x})$, the posterior probability of the model parameters given the data (Gopalan *et al*, 2015). To enable inference, we define a conditionally conjugate version of the model with an additional layer of latent variables. For each cell $i$ and gene $g$, we add $K$ latent variables $z_{i,g,k} \sim \text{Poisson}(\theta_{i,k}\beta_{g,k})$ such that $x_{i,g} = \Sigma_k z_{i,g,k}$. Because the sum of independent Poisson random variables is a Poisson random variable

with rate equal to the sum or the component rates, this alternative model preserves the marginal distribution of observed molecular counts. In the context of scRNA-seq, the auxiliary variables assign each observed molecule to a factor and can be thought of as the contribution of each factor to the observed molecular count $x_{i,g}$.

Under the augmented model, we posit a mean-field variational family over the latent variables:

$$q(\boldsymbol{\theta}, \boldsymbol{\beta}, \boldsymbol{\xi}, \boldsymbol{\eta}, \boldsymbol{z}) = \prod_{i,k} q(\theta_{i,k}|\gamma_{i,k}) \prod_{g,k} q(\beta_{g,k}|\lambda_{g,k}) \prod_i q(\xi_i|\kappa_i)$$
$$\prod_g q(\eta_g|\tau_g) \prod_{i,g} q(z_{i,g}|\phi_{i,g})$$

We set variational parameters to have the same form as their complete conditionals. Thus, $\gamma_{i,k}$, $\lambda_{g,k}$, $\kappa_i$, and $\tau_g$ are Gamma distributions with their own shapes and scales. $\varphi_{i,g}$ is a multinomial because the complete conditional of a bank of Poisson variables, given their sum, is a multinomial with a parameter proportional to the Poisson rates.

We fit the variational parameters to minimize the Kullback–Leibler (KL) divergence between the variational distribution and the true posterior using the algorithm described in Gopalan *et al* (2015), with some small modifications we have found helpful for scRNA-seq data. In the following optimization algorithm, we denote the shape and rate parameters of the variational approximation by the superscripts *shp* and *rte*, respectively. Our implementation terminates when the change to the marginal log likelihood is < 0.001 % twice in a row, checking every 10 iterations. Hyperparameters and initializations are described in the next section.

A. Set the shape parameters of the gene and cell capacities, where $K$ is the number of factors:

$$\kappa_i^{shp} = a' + Ka$$
$$\tau_g^{shp} = c' + Kc$$

B. Repeat until convergence:
   1  For each gene $g$, set the gene weights and capacity:

$$\lambda_{gk}^{shp} = c + \sum_i x_{ig}\phi_{igk}$$

$$\lambda_{gk}^{rte} = \frac{\tau_g^{shp}}{\tau_g^{rte}} + \sum_i \frac{\gamma_{ik}^{shp}}{\gamma_{ik}^{rte}}$$

$$\tau_g^{rte} = d' + \sum_k \frac{\lambda_{gk}^{shp}}{\lambda_{gk}^{rte}}$$

2 For each cell $i$, set the cell weights and capacity:

$$\gamma_{ik}^{shp} = a + \sum_g x_{ig} \phi_{igk}$$

$$\gamma_{ik}^{rte} = \frac{\kappa_i^{shp}}{\kappa_i^{rte}} + \sum_g \frac{\lambda_{gk}^{shp}}{\lambda_{gk}^{rte}}$$

$$\tau_g^{rte} = c' + \sum_k \frac{\gamma_{ik}^{shp}}{\gamma_{ik}^{rte}}$$

3 For each cell $i$ and gene $g$ such that $x_{i,g} > 0$, set the multinomial:

$$\phi_{ig} \propto \exp\{\Psi(\gamma_{ik}^{shp}) - \log \gamma_{ik}^{rte} + \Psi(\lambda_{gk}^{shp}) - \log \lambda_{gk}^{rte}\}$$

For scRNA-seq data, we have found that $\varphi_{ig}$'s update order (relative to the other variational parameters) can affect symmetry breaking. In particular, performing (3) as the first step of the first iteration (before (1) and (2)) can result in redundant factors with similar weights across cells and genes.

After optimizing the variational parameters, we use the variational distribution as a proxy for the posterior $p(\xi, \theta, \eta, \beta | x)$ in downstream analysis. For example, we use the variational approximation's means to calculate the cell and gene scores defined in the previous section. We can also work with the distributions directly, such as when we sample from them to perform the posterior predictive checks in Fig 2B and Appendix Fig S1.

### Hyperparameters and initialization

Hyperparameters $a'$, $b'$, $c'$, and $d'$ are set to preserve the empirical variance-to-mean ratio of the total molecules per cell or gene in the Gamma distributions from which $\xi$ and $\eta$ are drawn. Specifically, we set

$$b' = a' * \frac{E[\text{molecules per cell}]}{\text{Var}[\text{molecules per cell}]}$$

and

$$d' = c' * \frac{E[\text{molecules per gene}]}{\text{Var}[\text{molecules per gene}]}.$$

To preserve sparsity, we fix $a$ and $c$ to 0.3 and $a'$ and $c'$ to 1. In this scheme, we find the algorithm largely insensitive to small changes in the hyperparameters.

We initialize the variational distributions for $\xi$, $\theta$, $\eta$, $\beta$ to their priors times a random multiplier between 0.5 and 1.5. For each cell $i$ and gene $g$ such that $x_{i,g} > 0$, we initialize the (normalized) multinomial $\varphi_{ig}$ from a symmetric Dirichlet.

We note that, in original HPF paper, the rate hyperpriors for capacities $\xi$ and $\eta$ were defined as $a'/b'$ and $c'/d'$, whereas we define

them directly as $b'$ and $d'$. Those who wish to use the original notation while preserving the empirical variance-to-mean ratio should invert the fractions above when setting $b'$ and $d'$.

### Scalable inference on sparse scRNA-seq matrices

Because the likelihood of observed data under scHPF depends only on non-zero expression values, we only need to consider non-zero entries during training (Gopalan *et al*, 2015). This facilitates fast, memory-efficient inference on sparse scRNA-seq dataset. Training scHPF has $O(NK + MK + TK)$ computational complexity, where $N$ is the number of cells, $M$ is the number of genes, $K$ is the number of factors and $T$ is the number of non-zero matrix entries. In typical scRNA-seq datasets, $TK$ is the dominant term but is still much smaller than $NM$ for reasonable values of $K$. In theory, this gives scHPF a computational advantage over methods which must iterate through (and may also store) all $NM$ matrix entries.

### Selection of number of factors

In actually usage, such as the for the high-grade glioma demonstration in this paper, we select the number of factors $K$ such that (i) the model's log likelihood has converged (Appendix Fig S8A) and (ii) each well-defined cell type in the dataset is most strongly associated with at least one factor with which no other cell type is most strongly associated (Appendix Fig S8B–D). For benchmarking experiments, to avoid biasing results toward any one method, we set the number for factors to the smallest multiple of five greater than the number of clusters for the PBMC and Matcovitch *et al's* datasets, and to five for TS543 (Appendix Table S1). However, predictive performance was robust to a range of values for $K$ (Fig EV2).

### Normalization for benchmarking

Log-normalization was applied by adding 1 to molecular counts and then taking the logarithm, base 2. Counts per median (rate normalization) was calculated by normalizing the molecular counts in each cell to sum to 1 and then multiplying all values by the median number of molecules per cell. For log-rate normalization, we performed the log-normalization procedure described above on rate-normalized data.

### Other factorization methods

- We applied PCA, NMF, and FA using the scikit-learn python package, with default parameters (Pedregosa *et al*, 2011). Methods were tested on log, rate, and log-rate-transformed data.
- ZIFA was cloned from https://github.com/epierson9/ZIFA. To fit the model, we used the block_ZIFA implementation with parameter p0_thresh = 1 and otherwise default settings. Per its authors' specifications, we applied ZIFA to log- and log-rate-normalized data only.
- We ZINB-WaVe applied using the zinbFit function from the zinbwave R package. In accordance with the default parameter values, we included both cell and gene intercept terms. ZINB-WaVe was run on unnormalized count data.

### Benchmarking procedure

Prior to training, we randomly selected 4% of non-zero expression values to use as a held-out test set and 2% as a validation set.

The remaining data were used as a training set. By holding out only a small portion of data, we aimed to minimally impact datasets' native sparsity structure. As these test and validation sets were small compared to the training set, we evaluated methods' predictive performance on at least three randomly chosen partitions of the data into training, validation, and test sets. We ran each method-normalization pair with ten random initializations on each training set and selected the run with the lowest mean absolute error on the corresponding validation set. Due to ZIFA's long runtime (~ 23 h per initialization on TS543), we only ran it with one initialization per training set and for only one value of $K$. Similarly, ZINB-WaVE's high computational resource requirements precluded running it on both TS543 and on more than one value of $K$ for other datasets.

### Posterior predictive checks

We generated posterior predictive samples from scHPF by sampling latent representations $\theta_i$ and $\beta_g$ from the variational posterior and using their inner product as the rate of a Poisson, from which we sampled counts. For PCA, FA, and ZIFA, we sampled latent representations and expression values according to their underlying generative models (Bishop, 2006). For each method, normalization, and dataset, we sampled ten $N \times M$ datasets. Samples from models on normalized data were inverse transformed back to molecular counts before calculating column and row coefficients of variation. For example, samples from PCA on log-normalized data were added to −1 and then exponentiated (base 2) before calculating coefficients of variation. Each gene and cell's coefficients of variation were averaged across ten replicate posterior predictive simulations. The Kolmogorov–Smirnov test statistic was calculated using the python package scipy.

### Estimating memory consumption

We estimated methods' peak memory consumption using the Linux top utility.

### Preparation of TS543 glioma neurospheres

TS543 cells were plated at density $1 \times 10^4$ viable cells/cm$^2$ and grown as neurospheres with NeuroCult™ NS-A basal medium supplemented with NeuroCult™ NS-A proliferation supplement, 20 ng/ml EGF, 10 ng/ml bFGF, and 0.0002% heparin (Stem Cell Technologies). When diameters of neurospheres reached to approximately 100 μm, neurospheres were dissociated to single cells with mechanical force by pipetting 30–50 times.

### Radiographically guided biopsies of high-grade glioma

Human glioma surgical specimens were procured from de-identified patients who provided written informed consent to participate in these studies through a protocol approved by the Columbia Institutional Review Board (IRB-AAAJ6163). Radiographically guided biopsies were obtained as described in Gill *et al* (2014). Briefly, the patient studied here presented with FLAIR hyperintense, non-contrast-enhancing tissue along the surgical trajectory based on MRI between the craniotomy site and gadolinium contrast-enhancing border of the lesion. This region was biopsied and comprised the tumor margin specimen described above. A region of the contrast-enhancing core of the lesion was also biopsied and comprised the tumor core specimen.

### Whole-genome sequencing

Low-pass whole-genome sequencing (WGS) was conducted as described in Yuan *et al* (2018). Briefly, we homogenized tissue with a Dounce and extracted DNA and RNA with a ZR-Duet Kit (Zymo) according to the manufacturer's instructions. For the normal control, DNA and RNA were extracted using the same kit from peripheral blood mononuclear cells. WGS libraries were constructed by *in vitro* transposition using the Illumina Nextera XT kit and sequenced on an Illumina NextSeq 500 with $2 \times 75$ base paired-end reads to a depth of ~ 1×. Reads were aligned to the hg19 build of the human genome using *bwa-mem,* and the coverage for each chromosome was quantified using *bedtools* after collapsing PCR duplicates with *samtools*. To generate the bulk WGS heatmap in Appendix Fig S3E, we divided the normalized coverage of each chromosome in the tumor sample by that of the normal sample, normalized the resulting ratio by the median ratio across all chromosomes, and multiplied by two to estimate average copy number of each chromosome in the tumor sample. Note that we do not have consent to share the raw WGS data from these patients.

### Microwell-based scRNA-seq

Single-cell RNA-seq for TS543 and HGG samples was conducted as described in Yuan *et al* (2018) using a microwell array-based platform (Yuan & Sims, 2016).

- Freshly dissociated cells were live stained (Calcein AM, C3100MP, Thermo Fisher Scientific) on ice for 15–30 min.
- The stained cells (500 cells/μl) were then pipetted into a microwell array device. The cells were allowed to settle into the microwells for 3 min. Any un-trapped cells were flushed out with TBS (Tris-buffered saline, T5912, Sigma) buffer.
- Barcoded mRNA capture beads (500 beads/μl) were then loaded into the microwells followed by a TBS buffer flush.
- The cell and bead-loaded device was then connected to a computer-controlled reagent delivery and temperature control system. Lysis buffer [1% 2-mercaptoethanol (BP176-100, Fisher Scientific), 99% buffer TCL (1031576, Qiagen)] and perfluorinated oil (F3556-25ML, Sigma-Aldrich) were infused through the device in rapid succession to physically isolate individual microwells and lyse the trapped cells. The device was kept at 50°C for 20 min to further promote cell lysis and then at 25°C for 90 min for mRNA capture.
- Wash buffer supplemented with RNase inhibitor [0.02 U/μl SUPERaseIN (AM2696, Thermo Fisher Scientific) in wash buffer (20 mM Tris–HCl pH 7.9, 50 mM NaCl, 0.1% Tween-20)] was then flushed through the device to unseal the microwells and remove any uncaptured mRNA molecules. Reverse transcription mixture (1× Maxima RT buffer, 1 mM dNTPs, 1 U/μl SUPERaseIN, 2.5 μM template switch oligo, 10 U/μl Maxima H Minus reverse transcriptase (EP0752, Thermo Fisher Scientific), 0.1% Tween-20) was infused into the device followed by incubation at 25°C for 30 min and then at 42°C for 90 min.
- The beads were then extracted from the device and washed sequentially with TE/SDS buffer (10 mM Tris–HCl, 1 mM EDTA, 0.5% SDS), TE/TW buffer (10 mM Tris–HCl, 1 mM EDTA, 0.01% Tween-20, pH 8.0), and nuclease-free water. The beads were then treated with Exonuclease I reaction mixture [1× Exo-I buffer, 1 U/μl Exo-I (M0293L, New England Biolabs)] at 37°C for 30 min and

then washed sequentially with TE/SDS buffer (10 mM Tris–HCl, 1 mM EDTA, 0.5% SDS), TE/TW buffer (10 mM Tris–HCl, 1 mM EDTA, 0.01% Tween-20, pH 8.0), nuclease-free water and split into multiple 50 µl PCRs (1× Hifi Hot Start Ready mix (KK2601, Kapa Biosystems), 1 µM SMRTpcr primer). Twelve amplification cycles [95°C 3 min, four cycles of (98°C 20 s, 65°C 45 s, 72°C 3 min), eight cycles of (98°C 20 s, 67°C 20 s, 72°C 3 min), 72°C 5 min] were performed on a thermocycler. PCR product was pooled and purified using SPRI paramagnetic bead technology (Ampure, Beckman) with a bead-to-sample volume ratio of 0.6:1.

- Purified cDNA was then tagmented and further amplified using the Nextera kit for *in vitro* transposition (FC-131-1024, Illumina) with 0.6 ng cDNA used as input. The i5 index primer is replaced by a custom P5 Nextera PCR primer for the selective amplification of 5′ end of cDNA (corresponding to the 3′ end of mRNA). Two rounds of SPRI paramagnetic bead-based purification (Ampure, Beckman) with a bead-to-sample volume ratio of 0.6:1 and 1:1, respectively, were performed sequentially on the Nextera PCR product to obtain sequencing-ready libraries.

- The resulting single-cell RNA-seq libraries were spiked with 20% PhiX library (FC-110-3001, Illumina) and sequenced on a sequencer (NextSeq 500, Illumina). A custom read 1 sequencing primer was used.

### scRNA-seq data preprocessing

Reads for TS543 and HGG samples were aligned using STAR and processed into molecular count matrices as described in Yuan *et al* (2018). For all benchmarking and scHPF analyses, we only considered protein-coding genes that were expressed in at least 0.1% of cells in the dataset, rounded to the next largest multiple of 5 (Appendix Table S1).

### Identification of malignant glioma cells

We identified malignantly transformed cells by two orthogonal methods. First, we clustered cells' scRNA-seq profiles (see Clustering and visualization) and defined putative malignant cells using the genes most specific to each cluster (Appendix Figs S2 and S3A). Next, we performed PCA of cells' whole-chromosome expression and found that the first principal component, which we call the malignancy score, separated putatively transformed cells from non-malignant cells (Appendix Fig S3B–D). For further validation, we computed putative glioma cells' average chromosomal expression profiles relative to putative non-malignant cells and found that they were in good agreement with aneuploidies identified by low-coverage whole-genome sequencing of bulk tissue from the tumor core (Appendix Fig S5E).

### Clustering and visualization

Clustering, visualization, and identification of cluster-specific genes were performed similarly to (Yuan *et al*, 2018), with an updated method for selecting genes detected in fewer cells than expected given their apparent expression level (likely markers of cellular subpopulations). Briefly, for variable gene selection only, we normalized the molecular counts for each cell to sum to 1. Genes were then ordered by their normalized expression values. For each gene $g$, we calculated $f_g$, the fraction of cells in the dataset that expressed $g$, and $f_g^{max}$, the maximum $f_g$ in a rolling window of 25 genes centered on $g$. $f_g^{max}$ approximates the fraction of cells in which we expect to observe

transcripts given $g$'s overall expression in the dataset. The scaled difference between $f_g$ and $f_g^{max}$ defines $g$'s dropout score:

$$\text{dropout\_score}_g = \frac{|f_g - f_g^{max}|}{\sqrt{f_g^{max}}}.$$

We selected marker genes with dropout scores that are either > 0.15 or at least six standard deviations above the mean, inclusively.

To cluster and visualize the data, we computed a cell by cell Spearman's correlation matrix using the marker genes identified above. Using this matrix, we constructed a $k$-nearest neighbors graph ($k = 20$), which we then used as input to Louvain clustering with Phenograph (Levine *et al*, 2015). After clustering, we identified genes most specific to each cluster using a binomial test (Shekhar *et al*, 2016). The same similarity matrix, transformed into a distance matrix by subtracting its values from 1, was used as input to tSNE for visualization.

### Regional biases

$P$-values for both factors and top-scoring genes in each factor were calculated using the Mann–Whitney $U$-test and Bonferroni corrected for the total number of factors.

### Survival analysis

TCGA data for glioblastoma were downloaded from GDAC Firehose. Normalized expression values were $\log_2(RSEM + 1)$-transformed, and each factor's expression program was defined as its 25 highest scoring genes. We then calculated each program's mean relative expression for each donor, and $z$-scored these values across donors. For each program, donors with $z$-scores greater than a threshold $t$ were considered enriched, and all others were defined as not enriched. Patients with $z$-scores $< -t$ were considered depleted. For Fig 4G and H, we set $t = 1.5$. Appendix Fig S6 shows the analysis with a range of threshold values. Kaplan–Meier curves and log-rank test $P$-values were generated with the Lifelines v0.11.1 Python module.

# Data availability

The datasets produced in this study are available in the following database: scRNA-seq data: Gene Expression Omnibus GSE116621 (https://www.ncbi.nlm.nih.gov/geo/query/acc.cgi?acc=GSE116621). Code is available at: https://github.com/simslab/scHPF.

Expanded View for this article is available online.

## Acknowledgements

We thank the Sulzberger Columbia Genome Center for assistance and resources for high-throughput sequencing. P.A.S. was supported by NIH/NCI grant R33CA202827, NIH/NIBIB grant K01EB016071, NIH/NCI grant U54CA209997, and a Human Cell Atlas Pilot Project grant from the Chan Zuckerberg Initiative. P.A.S., A.I., and A.L. are supported by NIH/NCI grant U54CA193313. P.A.S., P.C., and J.N.B. are supported by NIH/NINDS grant R01NS103473. DB is supported by ONR 133691-5102004, NIH 5100481-5500001084, and the John Simon Guggenheim Foundation. F.J.R.R. is

supported by the EU Horizon 2020 program (Marie Sklodowska-Curie Individual Fellowship, grant agreement 706760).

## Author contributions

HML and DMB conceived of the method. HML, DMB, and PAS designed the study. JNB and PC procured glioma specimens. AL and AI prepared glioma samples. JY and YLC performed single-cell sequencing. HML and FJRR wrote code. HML and PAS analyzed data. ECB performed whole-genome sequencing. HML and PAS wrote the manuscript with input from DMB, PC, and FJRR. All authors read and approved the manuscript.

## Conflict of interest

J.Y. and P.A.S. are listed as inventors on patent applications filed by Columbia University related to the microwell technology described here for single-cell RNA-seq.

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
