## [Review Process File · Molecular Systems Biology]

De novo Gene Signature Identification from Single-Cell RNA-Seq with Hierarchical Poisson Factorization

Hanna Mendes Levitin, Jinzhou Yuan, Yim Ling Cheng, Francisco J.R. Ruiz, Erin C. Bush, Jeffrey N. Bruce, Peter Canoll, Antonio Iavarone, Anna Lasorella, David M. Blei, Peter A. Sims

Review timeline:

Submission date:	16 July 2018
Editorial Decision:	20 August 2018
Revision received:	16 November 2018
Editorial Decision:	18 December 2018
Revision received:	15 January 2019
Accepted:	21 January 2019

Editor: Maria Polychronidou

Transaction Report:

1st Editorial Decision

20 August 2018

Thank you again for submitting your work to Molecular Systems Biology. We have now heard back from the two referees who agreed to evaluate your study. As you will see below, the reviewers think that the presented method seems potentially useful for the analysis of single-cell RNA-seq data. They raise however a series of concerns, which we would ask you to address in a major revision.

Some of the more fundamental issues raised are the following:

- Reviewer #1 mentions that a more extensive comparison to existing approaches needs to be included. During our pre-decision cross-commenting process (in which the reviewers are given the chance to make additional comments, including on each other's reports), this reviewer emphasized that demonstrating concrete advantages over state-of-the-art existing approaches is particularly important.
- Reviewer #2 strongly suggests providing the software in a form that is easy to install and use, so that it can be easily adopted by others.
- Reviewer #2 also refers to the need to address a series of technical concerns.

All other issues raised need to be convincingly addressed. Please feel free to contact me in case you would like to discuss in further detail any of the issues raised by the reviewers.

REFeree REPORTS

Reviewer #1:

Levitin et al describe a novel method for the identification of gene modules from single-cell RNA-seq data. Their method is based on an adaptation of Poisson factorization and does not require prior

normalization. They have compared their approach to PCA, NMF, factor analysis and ZIFA. Additionally, they have applied their approach to scRNA-seq of a glioma cell line, to demonstrate its use.

The method developed is novel. However, the availability of many competing methods with similar performances diminish the interest in the authors' approach. I would ask the authors to consider the following:

1. scHPF has not been vetted against all relevant competing methods. In particular, ZINB (<https://www.nature.com/articles/s41467-017-02554-5>) provides the functionality of the authors approach and should be compared to their method. Likewise scvis (<https://www.nature.com/articles/s41467-018-04368-5>) should also be compared. There are actually many, many other methods for dimensionality reduction and I ask the authors to at least compare to these two using some of the tests that are described in those papers. A complete evaluation would include some of the best of other competing approaches.
2. The comparisons that the authors make to other methods are not adequate. One of the problems with PCA and other approaches is that the first component often correlates with some technical variable (such as number reads, #aligned, duplication rate, etc.) The authors should correlate the first 1-2 components of the reduced representation of their data to such technical variables and demonstrate superior performance to competing methods.
3. Following up on this, the authors should assess the ability of scHPF to recover the true underlying low-dimensional signal and clustering structure based on simulated and/or experimental data. See Figure 7 of the ZINB paper and Figure 2 of the scvis papers for examples.
4. The only comparison of their novel method to competing methods is done in their Figure 2. However, the results of this comparison are not compelling as competing methods seem to have similar performances.
5. The held-out test set seems somewhat arbitrary. I understand the rationale for holding out such a small percentage as test (4%), but it seems that this procedure should then be repeated for different subsets of 4%. This would yield error bars for Figure 2A and clear up question 4.
6. The survival analysis is not satisfactory. Typically, above median and below median expression is used to divide subjects into groups. The 1.5 and -1.5 thresholds seem arbitrary, ignore a large portion of the population and seem cherry-picked to produce the desired outcome.
7. Other approaches for identifying useful gene modules have not been considered as competing approaches, such as single-cell co-expression network clustering.

Reviewer #2:

Levitin et al. present scHPF, an updated version of HPF tailored specifically to single-cell RNA-seq data.

Let me first say that you have picked exactly the right reviewer. I am likely the only one on Earth who has personally studied nearly every line of code of the original HPF C++ implementation, personally implemented a Python version of the algorithm, and tested it on a wide range of single-cell RNA-seq datasets. Yet I have had no prior interaction with the authors on this subject.

Needless to say, I think HPF (and therefore scHPF) is a highly significant advance for scRNA-seq. It solves several challenges:

- * It decomposes scRNA-seq datasets into highly interpretable components
- * Both cells and genes are allowed to be (potentially) composites of multiple components, in agreement with the observation that genes participate in different gene-modules depending on context, and that cell identity is often modular
- * It uses arguably the right noise model (gamma-poisson mixtures, equivalent to the negative binomial), resulting in very accurate fits to real data
- * Empirically, it is able to compress a large amount of information into a small number of components. E.g. from 2000 genes to 64 components with very little loss of information (as judged by posterior predictive samples)
- * It operates on sparse matrices and converges rapidly, so that it can be applied to today's very large scRNA-seq datasets (unlike some competing algorithms)

scHPF will have many important applications, including

- * As dimensionality reduction before e.g. KNN graph construction, or tSNE projection
- * To discover functional gene modules
- * For separation of distinct aspects of cellular dynamics prior to lineage inference (e.g. cell cycle from maturation)
- * Potentially to remove components related to technical artefacts (e.g. IEG expression or sex-specific gene expression)

The present manuscript adds only minor algorithmic improvements over the original HPF paper. But beyond that it adds:

- * A Python implementation on top of TensorFlow (for parallel computation and potentially exploiting GPUs)
- * Extensive example applications to real-world scRNA-seq datasets

I have a few minor issues with the presentation, some more significant concerns about the code, and some suggestions for improvements or additional applications:

1. I think the argument for why the gamma-poisson is a suitable model for scRNA-seq data (assuming UMIs) is strong. You may want to cite [Comparative Analysis of Single-Cell RNA Sequencing Methods - ScienceDirect](<https://www.sciencedirect.com/science/article/pii/S1097276517300497>) (Fig. 5B) as well as some of the papers cited in the first part of the Result section in [K-nearest neighbor smoothing for high-throughput single-cell RNA-Seq data | bioRxiv](<https://www.biorxiv.org/content/early/2018/04/09/217737>).
2. "We generated posterior predictive samples from scHPF by sampling latent representations θ and β from the variational posterior and taking the inner product." but this results in real numbers, not integers. I assume you then drew a Poisson sample?
3. In heterogeneous scRNA-seq data, consisting of disparate cell types, zeros are often inflated (simply because the genes are not expressed in all cell types). HPF accounts nicely for this due to the sparsity of the components. However, I think you should also account for it when plotting: instead of using a linear color scale, zeros should be plotted qualitatively differently. This greatly helps interpret the plots. I like to plot zeros in very light gray (so that the cells are visible but unobtrusive) and non-zeros on a linear scale using the perceptually uniform viridis scale. Here's a slightly hacky way to generate such a color scale:

```
import matplotlib.colors as colors
import matplotlib.pyplot as plt
zviridis = colors.LinearSegmentedColormap.from_list("zviridis", [(0.9, 0.9, 0.9, 1)] +
list(plt.cm.viridis(np.arange(1000) / 1000)), N=1001)
```

4. The code is currently quite messy. There is a mixture of unrelated concerns: file I/O, feature selection, training, etc. which makes it hard to use the algorithm in any other way than as a command-line tool. I would strongly suggest isolating the algorithm itself into a class that roughly follows the scikit-learn API (which can be as simple as having a `fit()` method). That would make it much easier to integrate scHPF in software pipelines.
5. It would be nice to have a pip installable package that pulls in all the requirements (dependencies). pip can also be configured to automatically create and make available a command-line tool, so that you could run "schpf" from anywhere.
6. The documentation says that the input file should be whitespace-delimited, but the code requires tab-delimited. It also doesn't allow a trailing tab. Generally speaking, ad-hoc file formats are very error-prone and force users to spend time writing file generators and parsers, which is super-boring.
7. It would be better if the user could supply data in some well-defined format. I would suggest allowing loom files as input and to store the result back in the same loom file, in the form of attributes (e.g. "theta" on the cells). Loom allows attributes to be 2D matrices. Loom files are compatible with many popular pipelines including Seurat, scope and scanpy. See loompy.org.
8. Did you try using the HPF factors as input to tSNE? In my experience, that works really well, and much better than using the raw matrix (obviously) or the commonly used top PCA components (less obviously). One possible advantage over your current approach is that the tSNE becomes independent of clustering.
9. Can a fitted HPF model be used to project additional data into the latent space? This is useful in many situations, for example for RNA velocity when there is a need to project extrapolated cell states. In my own HPF implementation, I achieved this by holding β fixed while learning a new

theta from the new data. This works, but I'm not sure if it's technically sound.

10. Validation and test data is not used in the ordinary sense. Instead, they represent held-out datapoints (subsets of the training data that are set to zero). Doesn't this affect the fitting? I.e. if a large fraction (say 50%) of the input data is used for validation, then the training data will be distorted. I guess this is fine, but should maybe be pointed out, and you might want to give a proposal as to the fractions that are suitable for test and validation.

11. One way that HPF gets stuck in a local maximum, is by merging factors. In other words, it tends to find multiple near-identical factors, with near-identical beta and theta matrices. I suppose once two factors merge, they cannot diverge because locally they are already optimal. The result is in effect a reduction in the total number of factors, which may be fine of course. In my own code, I tried identifying redundant factors, and reinitializing them to a random state, but this did not help. Are redundant factors an indication that there is no more variance to be found?

12. The hyperparameters are a bit confusingly named. For example, *c* is used to index the cells, and is also a hyperparameter, distinct from *c'*. I realize this is how it was done in the HPF paper, so maybe not a good idea to change now, but it would be helpful if you point out that *c* has double meaning.

13. Interestingly, your TensorFlow version is ~25% slower than my pure-python (numpy) version, even though TensorFlow uses all four of my cores, whereas numpy uses only one. It may be worth investigating if e.g. the datatypes matter (float64 vs float32 or even float16), if there's excessive copying of arrays somewhere, or something else.

In summary, I am very much in favor of publication, but to maximize impact, the software should be re-architected to be easy to install and use, both as a command-line tool and as a component in a larger pipeline.

1st Revision - authors' response

16 November 2018

Response to Reviewer Comments for MSB-18-8557, *De novo Gene Signature Identification from Single-Cell RNA-Seq with Hierarchical Poisson Factorization*

Reviewer comments appear below in *italics* with point-by-point responses. Modifications to the manuscript appear in **purple** text below and in the revised manuscript.

REVIEWER 1

Levitin et al describe a novel method for the identification of gene modules from single-cell RNA-seq data. Their method is based on an adaptation of Poisson factorization and does not require prior normalization. They have compared their approach to PCA, NMF, factor analysis and ZIFA. Additionally, they have applied their approach to scRNA-seq of a glioma cell line, to demonstrate its use.

The method developed is novel. However, the availability of many competing methods with similar performances diminish the interest in the authors' approach.

We agree that there are many competing methods, and we have improved the manuscript in response to specific comments below to better highlight the unique advantages of scHPF. Importantly, the much of the paper focuses on the application of scHPF to scRNA-seq of radiographically localized human glioma surgical specimens taken from different regions of the same tumor. So, the example application provided in this study is considerably more sophisticated and relevant than “scRNA-seq of a glioma cell line”.

- 1) *scHPF has not been vetted against all relevant competing methods. In particular, ZINB (<https://www.nature.com/articles/s41467-017-02554-5>) provides the functionality of the authors approach and should be compared to their method. Likewise scvis (<https://www.nature.com/articles/s41467-018-04368-5>) should also be compared. There are actually many, many other methods for dimensionality reduction and I ask the authors to at least compare to these two using some of the tests that are described in those papers. A complete evaluation would include some of the best of other competing approaches.*

We have conducted a formal benchmarking analysis of scHPF against the two methods suggested by the reviewer (ZINB-WaVE and scvis). scHPF outperforms both methods on

multiple fronts as detailed below and in the revised manuscript. However, based on the manuscripts reporting these two techniques, neither is actually a directly “competing method”. scHPF is designed for *de novo* gene signature identification, which means that it takes genome-wide raw count data as input (to the extent that is possible given the limited sensitivity of scRNA-seq, typically >10,000 genes) and attempts to factorize it into the major transcriptional programs underlying a cellular population. This task is not demonstrated in any of the examples shown in the report of ZINB-WaVE (Risso et al, *Nature Communications*, 2018). Instead, the authors first identify the top ~1,000 highly variable genes in the data by external means (which are likely representative of subpopulation markers) followed by ZINB-WaVE dimensionality reduction. This means that no weight is assigned for any of the resulting components for genes that do not meet this initial variability criterion (~95% of genes). As we will describe below, when you actually compare the performance of ZINB-WaVE to that of scHPF on a much larger data matrix (filtered only to exclude genes expressed in (~0.1% of cells), ZINB-WaVE gives very inconsistent results, fails to recapitulate basic clustering of highly dissimilar cellular lineages, and is prohibitively computationally expensive. The case of scvis is even more extreme, because scvis is principally a visualization tool that is optimized, as the authors state in their abstract, for producing “interpretable two-dimensional representations of high-dimensional single-cell RNA-sequencing data”. So there is no expectation that scvis is going to take genome-wide data as input and produce higher dimensional decompositions (e.g. >2-factor) of scRNA-seq data like scHPF.

To benchmark the predictive performance of ZINB-WaVE and scvis on held-out test data, we used the procedure described in our original submission across 2-3 datasets. In the original submission, we tested PCA, NMF, FA, ZIFA, and scHPF with three different normalization methods on a >4,000-cell PBMC data set from 10x Genomics, a >3,000-cell MARS-seq microglial dataset from Matcovitch *et al*, and a ~10,000-cell glioma neurosphere dataset that we generated internally on our microwell array platform. While we tested scvis with all three datasets and normalization methods, we tested ZINB-WaVE without normalization (the most appropriate input type for both scHPF and ZINB-WaVE) only with the first two datasets due to prohibitive computing costs. As shown in **Fig. 2a** of the revised manuscript, ZINB-WaVE slightly outperforms scHPF on the PBMC dataset and very substantially under-performs scHPF on the microglial dataset. In fact, on the microglial dataset, ZINB-WaVE performs worse than any of the methods we tested on any dataset in terms of predictive performance. We repeated this same test for scvis across all three datasets and normalization methods, and found that scHPF significantly outperformed scvis in all cases (see figure below).

Because ZINB-WaVE slightly outperformed scHPF on the PBMC dataset, we decided to more carefully examine the resulting factors. While ZINB-WaVE exhibited good predictive performance, this does not necessarily translate into an interpretable representation of the underlying biology. We first applied a conventional clustering pipeline to the PBMC dataset (identification of highly variable genes followed by Louvain community detection) and, as expected, identified all of the major PBMC types including monocytes, dendritic cells, T cells, and B cells. As shown in **Fig. EV3** of the revised manuscript, the factors obtained using scHPF are in excellent agreement with the conventional clustering results. Indeed, each of the major clusters has an associated dominant factor, and there are relationships between the factors associated with related cell types (like natural killer cells and effector memory CD8 T cells or among the three monocyte/myeloid lineages represented). Conversely, the factors obtained using ZINB-WaVE do not exhibit the same close relationship to the basic cell types in the data. While there are dominant factors for monocytes and B cells, the smaller clusters do not relate to the ZINB-WaVE factors in a straightforward way.

Finally, we compared all of the methods to scHPF in terms of computational expense (**Table 1, copied below**). We found ZINB-WaVE to be extremely computationally expensive in terms of both computing time and memory usage compared to scHPF. Taken together with the issues described above and the fact that ZINB-WaVE was not really designed to perform the same computational task as scHPF, we feel that our work with scHPF is an important and unique contribution. As for scvis, we feel that its performance for this application is so poor that we would rather not show the results in our manuscript. We feel that it would be unfair to the authors of scvis who were very specific in their

original report as to the intended application of the algorithm, which is different from what we are doing here.

The above figure does not appear in the revision. A) is the mean squared error ratio between scvis and scHPF for predictive performance across three normalization conditions and three datasets. scHPF outperforms scvis in all cases. B) is the posterior predictive check results comparing the true CV distribution for cells and genes to the results of scHPF, ZIFA, factor analysis, PCA, and scvis. The performance of scvis is so poor that it is often difficult to plot the results on the same graph. For lograte normalization, we couldn't even obtain the final distribution due to numerical overflow errors. This is not a criticism of scvis, which is not meant for this application

	Parallelization	Runtime	Peak Memory
ZINB-WaVE	1 vCPU	442.58 min. (7.38 hours)	17.8 Gb
ZINB-WaVE	2 vCPU	237.76 min. (3.96 hours)	31.5 Gb
scHPF	2 vCPU	10.66 min.	1.6 Gb
scHPF	8 vCPU	6.13 min.	1.6 Gb
scHPF	16 vCPU	2.47 min	1.7 Gb
scHPF (10 trials)	8 vCPU	52.90 min.	1.6 Gb
scHPF (10 trials)	16 vCPU	36.45 min.	1.6 Gb

Table 1: Runtime and peak memory consumption at different levels of parallelization for ZINB-WaVE and scHPF on the PBMC dataset with $K=10$. ZINB-WaVE's high memory consumption precluded running it with more than 2 threads on this dataset.

Revisions to main text:

We compared scHPF's predictive performance to that of PCA, NMF and Factor Analysis (FA), as well as two methods developed specifically for scRNA-seq: Zero Inflated Factor Analysis (ZIFA) (Pierson & Yau, 2015) and Zero-Inflated Negative Binomial-based Wanted Variation Extraction (ZINB-WaVE) (Risso et al, 2018).

ZINB-WaVE was applied directly to molecular counts. We did not apply ZINB-WaVE to the nearly 10,000-cell TS543 dataset due to the method's prohibitive computational cost (Table 1). With only one exception, scHPF had the best predictive performance on held-out test data across all datasets and normalizations (Figure 2a). scHPF's superior performance was robust across a range of values for K , the number of factors (Extended Figure 2). Notably, while ZINB-WaVE had better predictive performance than scHPF on PBMCs, it had the highest mean squared error of any method on the Matcovitch *et al.* dataset.

Because scHPF and ZINB-WaVE performed comparably in terms of predictive performance on the PBMC dataset, we carefully examined their respective factorizations in terms of computational expense and biological interpretability. For a single initialization with $K = 10$, training ZINB-WaVE took 7.38 hours and had a peak memory consumption of 17.8 Gb (**Table 1**). Using two threads reduced ZINB-WaVE's runtime to just under four hours, but nearly doubled its memory consumption to 31.5 Gb. In contrast, our scHPF implementation took 2.5-10.7 minutes, depending on the number of threads available, and ~1.6 Gb of memory (**Table 1**). scHPF's superior performance is in part due to optimized compilation and automatic parallelization with the Python Numba library (Lam et al, 2015). In addition, unlike ZINB-WaVE, scHPF only needs to consider nonzero matrix entries during training (Methods), which imparts a considerable theoretical advantage over methods that must iterate through (and in some cases store) every matrix entry. Finally, we compared the interpretability of scHPF and ZINB-WaVE's low dimensional representations of cells in the PBMC data. Clustering using a conventional pipeline identified all of the major PBMC types including monocytes, dendritic cells, T cells, and B cells (Methods, **Fig EV3a**). scHPF factors were in excellent agreement with clustering results (**Figure EV3b,c**). Each major cell type had an associated dominant factor and there were relationships between factors associated with related cell types. In contrast, factors obtained using ZINB-WaVE did not exhibit the same close relationship to basic cell types in the data. While there were dominant factors for monocytes and B cells, smaller clusters did not relate to ZINB-WaVE factors in an interpretable way.

- 2) *The comparisons that the authors make to other methods are not adequate. One of the problems with PCA and other approaches is that the first component often correlates with some technical variable (such as number reads, #aligned, duplication rate, etc.) The authors should correlate the first 1-2 components of the reduced representation of their data to such technical variables and demonstrate superior performance to competing methods.*

We agree that the relationship between the scHPF factors and certain technical variables should be investigated. However, we disagree that correlation with technical variables is a disadvantage of PCA or any dimensionality reduction technique. In our opinion, the PCA's ability to identify technical variables that dominate the structure of a dataset is among its principal advantages and most common applications. Similarly, if we were to identify scHPF factors that correlate strongly with technical variables, we could eliminate those factors from downstream analysis to produce a less biased result. We assessed the correlation between the scHPF factors that we identified for the high-grade glioma (HGG) tissue data set and three technical variables of particular relevance to scRNA-seq – number of molecules detected per cell, number of genes detected per cell, and number of reads per molecule for each cell (duplication rate). We note that, unlike PCA where each component can be weighted and ranked by its eigenvalue, scHPF does not have a “first 1-2 components”, and so we made this assessment across all factors. We found that most factors were essentially uncorrelated with these technical variables and some factors had modest correlation (see **Appendix Fig. S4** in the revised manuscript). Importantly, correlation with technical variables like number of molecules or genes detected per cell does not always indicate a technical artifact in scRNA-seq. For example, we tend to detect more molecules and genes in cells that are physically larger or in clusters associated with multiplets. Interestingly, the two factors with modest correlation with number of molecules detected per cell corresponded to endothelial cells (most likely the largest cell type in the data set) and cell cycle control (enriched in mitotic figures on the verge of becoming two cells).

Revisions to text:

Cell's scHPF scores were largely uncorrelated with technical variables (**Appendix Figure S4**); however, two factors associated with physically larger cell types (dividing and endothelial) were modestly correlated with the number of molecules and genes per cell.

- 3) *Following up on this, the authors should assess the ability of scHPF to recover the true underlying low-dimensional signal and clustering structure based on simulated and/or experimental data. See Figure 7 of the ZINB paper and Figure 2 of the scvis papers for examples.*

We agree that it is important to assess the ability of scHPF to recover the true underlying low-dimensional signal (e.g. clustering structure). In the original manuscript, we investigated this extensively in **Fig. 3** and **Fig. S6 (now Fig. EV4)** for the glioma tissue data set. We showed a direct comparison between conventional clustering analysis of the data and the scHPF factors, which corresponded very well. In addition, we showed how the top-ranked genes in the scHPF factors corresponded to markers of the expected cell types in the glioma specimen (**Fig. S6, now EV4**). In the revised manuscript, we show additional examples of this including **Fig. EV3**, which compares the agreement between scHPF factors and conventional clustering for PBMCs to that obtained for ZINB-WaVE as described in greater detail above in response to 1).

Revisions to text:

Finally, we compared the interpretability of scHPF and ZINB-WaVE's low dimensional representations of cells in the PBMC data. Clustering using a conventional pipeline identified all of the major PBMC types including monocytes, dendritic cells, T cells, and B cells (Methods, **Fig EV3a**). scHPF factors were in excellent agreement with clustering results (**Figure EV3b,c**). Each major cell type had an associated dominant factor and there were relationships between factors associated with related cell types. In contrast, factors obtained using ZINB-WaVE did not exhibit the same close relationship to basic cell types in the data. While there were dominant factors for monocytes and B cells, smaller clusters did not relate to ZINB-WaVE factors in an interpretable way.

- 4) *The only comparison of their novel method to competing methods is done in their Figure 2. However, the results of this comparison are not compelling as competing methods seem to have similar performances.*

We respectfully disagree with this assessment and the claim that “competing methods seem to have similar performance” and that “the only comparison... is done in their Figure 2”. In **Fig. 2A** of the original manuscript, scHPF consistently outperforms competing methods, but typically only by ~20% in terms of the mean-squared error summary statistic for predictive performance. However, as shown in **Fig. 2B-C**, this result can be deceiving when it comes to posterior predictive checks of the underlying noise distribution (computed here as coefficient of variation or CV) both at the cell- and gene-levels. The noise distribution is of paramount importance in scRNA-seq, because it is the basis on which the major sources of variability in the underlying cellular population are identified by comparison to overall technical noise. Indeed, for the cell-level CV distribution, the KS-statistic (which assesses the dissimilarity between the true and predicted distributions) is 3-9-fold higher across all competing methods (**Fig. 2B, left**) with all samples included and 6-9-fold higher when negative samples are rectified (**Fig. 2C, left**). Similarly, for gene-level CV distributions, we find the KS-statistic to be 2-12-fold higher with all samples included for all methods except FA with log-rate normalization (**Fig. 2B, right**) and 3-29-fold higher with negative samples rectified for all methods (**Fig. 2C, right**). We think these large deviations for the posterior predictive checks and the consistent performance of scHPF make for a compelling comparison. We made further comparisons to competing methods in **Fig. S2,3 (now Fig. EV2 and Appendix Fig. 1)** with additional datasets and across many values of K in the supplement. Importantly, while we have added a figure showing the mean-squared error comparison for ZINB-WaVE to **Fig. 2A**, we could not produce the CV distribution comparisons for ZINB-WaVE because it is not a generative model. This is discussed in detail in response to comment 1).

- 5) *The held-out test set seems somewhat arbitrary. I understand the rationale for holding out such a small percentage as test (4%), but it seems that this procedure should then be repeated for different subsets of 4%. This would yield error bars for Figure 2A and clear up question 4.*

We agree that it would be valuable to provide error bars for **Fig. 2A** in the manner suggested by the reviewer. We now provide this information in **Fig. 2A** in the revised manuscript.

Revisions to text:

Error bars show standard error of the mean across three train/validation/test splits (Methods).

- 6) *The survival analysis is not satisfactory. Typically, above median and below median expression is used to divide subjects into groups. The 1.5 and -1.5 thresholds seem arbitrary, ignore a large portion of the population and seem cherry-picked to produce the desired outcome.*

We respectfully disagree that “typically, above median and below median expression is used to divide subjects in to groups”. We assert that this approach would preclude the analysis of many common and important phenotypic distributions in a population. For example, consider the case that the phenotype is overexpression of a gene-of-interest, and that this phenotype is relatively rare. Specifically, consider that its distribution across patients is $X = (0,0,1,1,1,1,1,1,1,1,2,2,2,1000,1000)$, where the patients with 1000 counts are of particular interest. The median of X is 1, and so the two groups would be $X_1 = (0,0,1,1,1,1,1,1,1,1)$ and $X_2 = (2,2,2,1000,1000)$, creating a situation where most of the subjects in the “high expression” group, X_2 , do not have the phenotype-of-interest. In reality, there is usually a competition between statistical power and the threshold chosen for effect size. Selecting a threshold such to isolate “elite” subjects with a very high effect size will typically result in lower statistical power. To demonstrate this principal and to provide a broader range of effect size thresholds to demonstrate the robustness of our findings, we generated **Appendix Fig. 6** for the revised manuscript which shows how the statistical significance of the originally reported survival difference varies with the effect size threshold:

At very high effect size thresholds ($t=1.8$), the survival difference between depleted and enriched groups becomes insignificant because the group sizes are too small for there to be sufficient statistical power. At very low effect size thresholds ($t=1.2$), we are no longer actually comparing subjects with the phenotype of interest to subjects without it. The threshold is low enough that many subjects in the “high expression” group actually exhibit low expression of the gene signature, and the survival difference is also not significant. At intermediate values of the threshold, we observe a significant survival difference, and at all values of the threshold, we observe a survival difference in the same direction. Importantly, for all thresholds tested, comparison of the high expression “enriched” group to the remaining patients gives a statistically significant survival difference.

Revisions to text:

Restricting the analysis to glioblastoma (GBM), we identified patients enriched and depleted for the top genes in each factor (Methods, **Appendix Figure S6** for analysis of sensitivity to effect size thresholds).

Appendix Figure S6: Kaplan-Meier curves show survival differences in TCGA for donors enriched (red), not enriched (purple), and depleted (blue) for the 25 top scoring genes in astrocyte-like factor 1 (Methods) at different effect size cutoffs. Median survival difference (Med. SD) increases as the effect size cutoff (t) for inclusion in enriched and depleted cohorts increases. Statistical power decreases as effect size increases and treatment groups become smaller.

- 7) *Other approaches for identifying useful gene modules have not been considered as competing approaches, such as single-cell co-expression network clustering.*

We agree that single-cell co-expression network clustering is an important class of analytical techniques particularly for inferring regulatory relationships between genes, but these methods really do not perform the same task as scHPF. Because of transcript drop-out and noise, the main issues that scHPF has been developed to overcome, co-expression network clustering is very technically challenging with scRNA-seq data as discussed in, for example, Crow et al, *Genome Biology*, 2016 and Crow and Gillis, *Trends in Genetics*, 2018. The scope of computational methods for analyzing scRNA-seq data is vast, and so

we limited our benchmarking and comparisons to methods we thought were most directly analogous to scHPF (i.e. matrix factorization methods).

REVIEWER 2

Levitin et al. present scHPF, an updated version of HPF tailored specifically to single-cell RNA-seq data.

Let me first say that you have picked exactly the right reviewer. I am likely the only one on Earth who has personally studied nearly every line of code of the original HPF C++ implementation, personally implemented a Python version of the algorithm, and tested it on a wide range of single-cell RNA-seq datasets. Yet I have had no prior interaction with the authors on this subject.

Needless to say, I think HPF (and therefore scHPF) is a highly significant advance for scRNA-seq. It solves several challenges:

- * It decomposes scRNA-seq datasets into highly interpretable components*
- * Both cells and genes are allowed to be (potentially) composites of multiple components, in agreement with the observation that genes participate in different gene-modules depending on context, and that cell identity is often modular*
- * It uses arguably the right noise model (gamma-poisson mixtures, equivalent to the negative binomial), resulting in very accurate fits to real data*
- * Empirically, it is able to compress a large amount of information into a small number of components. E.g. from 2000 genes to 64 components with very little loss of information (as judged by posterior predictive samples)*
- * It operates on sparse matrices and converges rapidly, so that it can be applied to today's very large scRNA-seq datasets (unlike some competing algorithms)*

scHPF will have many important applications, including

- * As dimensionality reduction before e.g. KNN graph construction, or tSNE projection*
- * To discover functional gene modules*
- * For separation of distinct aspects of cellular dynamics prior to lineage inference (e.g. cell cycle from maturation)*
- * Potentially to remove components related to technical artefacts (e.g. IEG expression or sex-specific gene expression)*

The present manuscript adds only minor algorithmic improvements over the original HPF paper. But beyond that it adds:

- * A Python implementation on top of TensorFlow (for parallel computation and potentially exploiting GPUs)*
- * Extensive example applications to real-world scRNA-seq datasets*

We agree with this synopsis and appreciate the encouraging comments.

- 1) I think the argument for why the gamma-poisson is a suitable model for scRNA-seq data (assuming UMIs) is strong. You may want to cite [Comparative Analysis of Single-Cell RNA Sequencing Methods - ScienceDirect] (<https://www.sciencedirect.com/science/article/pii/S1097276517300497>) (Fig. 5B) as well as some of the papers cited in the first part of the Result section in [K-nearest neighbor smoothing for high-throughput single-cell RNA-Seq data | bioRxiv] (<https://www.biorxiv.org/content/early/2018/04/09/217737>).*

We agree with this assessment and appreciate the suggested references, which we have added to the manuscript.

Revisions to text:

Previous work suggests that the Gamma-Poisson mixture distribution is an appropriate noise model for scRNA-seq data with unique molecular identifiers (UMIs) (Wagner et al, 2018; Ziegenhain et al, 2017).

- 2) "We generated posterior predictive samples from scHPF by sampling latent representations θ and β from the variational posterior and taking the inner*

product." but this results in real numbers, not integers. I assume you then drew a Poisson sample?

We appreciate your catching this mistake. Indeed, we did draw a Poisson sample and have modified the manuscript accordingly.

Revisions to text:

We generated posterior predictive samples from scHPF by sampling latent representations θ_i and β_g from the variational posterior and using their inner product as the rate of Poisson, from which we sampled counts.

- 3) *In heterogeneous scRNA-seq data, consisting of disparate cell types, zeros are often inflated (simply because the genes are not expressed in all cell types). HPF accounts nicely for this due to the sparsity of the components. However, I think you should also account for it when plotting: instead of using a linear color scale, zeros should be plotted qualitatively differently. This greatly helps interpret the plots. I like to plot zeros in very light gray (so that the cells are visible but unobtrusive) and non-zeros on a linear scale using the perceptually uniform viridis scale. Here's a slightly hacky way to generate such a color scale:*

```
import matplotlib.colors as colors
import matplotlib.pyplot as plt
zviridis = colors.LinearSegmentedColormap.from_list("zviridis", [(0.9, 0.9, 0.9, 1)] +
list(plt.cm.viridis(np.arange(1000) / 1000)), N=1001)
```

We understand your suggestion. However, the reason we did not do this is because the low values coloring the plots to which you refer are not actually zero and instead span a range of small values, and so we are hesitant to represent them all with a single color that is offset from the continuous colormap.

- 4) *The code is currently quite messy. There is a mixture of unrelated concerns: file I/O, feature selection, training, etc. which makes it hard to use the algorithm in any other way than as a command-line tool. I would strongly suggest isolating the algorithm itself into a class that roughly follows the scikit-learn API (which can be as simple as having a fit() method). That would make it much easier to integrate scHPF in software pipelines.*

We agree with this assessment and have made a significant effort to overhaul the code in terms of its organization, computational performance, ease-of-use, and easy-of-installation. This will be described in more detail below, but we have taken your specific suggestion of enabling integration following the scikit-learn API.

- 5) *It would be nice to have a pip installable package that pulls in all the requirements (dependencies). pip can also be configured to automatically create and make available a command-line tool, so that you could run "schpf" from anywhere.*

We completely agree with this suggestion and have created a pip-installable package.

- 6) *The documentation says that the input file should be whitespace-delimited, but the code requires tab-delimited. It also doesn't allow a trailing tab. Generally speaking, ad-hoc file formats are very error-prone and force users to spend time writing file generators and parsers, which is super-boring.*

We thank the reviewer for pointing out this discrepancy, which we have rectified in the new version of the code.

- 7) *It would be better if the user could supply data in some well-defined format. I would suggest allowing loom files as input and to store the result back in the same loom file, in the form of attributes (e.g. "theta" on the cells). Loom allows attributes to be 2D matrices. Loom files are compatible with many popular pipelines including Seurat, scope and scanpy. See loompy.org.*

We thank the reviewer for this suggestion and have enabled the use of loom files as input to preprocessing in the new version of the code.

- 8) *Did you try using the HPF factors as input to tSNE? In my experience, that works really well, and much better than using the raw matrix (obviously) or the commonly used top PCA components (less obviously). One possible advantage over your current approach is that the tSNE becomes independent of clustering.*

We agree – HPF factors do serve as highly effective input for both tSNE and UMAP as we now demonstrate in **Fig. EV5**.

Revisions to text:

In addition, we could use scHPF's factorization as a low-dimensional input to t-Distributed Stochastic Neighbor Embedding (t-SNE) (Maaten & Hinton, 2008) or Uniform Manifold Approximation and Projection (UMAP) (McInnes et al, 2018) to produce visualizations that were consistent with conventional clustering (**Figure EV5**). Taken together, these results show that scHPF captures the major features identified by standard analyses of this dataset.

- 9) *Can a fitted HPF model be used to project additional data into the latent space? This is useful in many situations, for example for RNA velocity when there is a need to project extrapolated cell states. In my own HPF implementation, I achieved this by holding beta fixed while learning a new theta from the new data. This works, but I'm not sure if it's technically sound.*

Yes – a fitted HPF model can be used to project additional data into the latent space as you describe. We have added this functionality in the new version of the code.

- 10) *Validation and test data is not used in the ordinary sense. Instead, they represent held-out datapoints (subsets of the training data that are set to zero). Doesn't this affect the fitting? I.e. if a large fraction (say 50%) of the input data is used for validation, then the training data will be distorted. I guess this is fine, but should maybe be pointed out, and you might want to give a proposal as to the fractions that are suitable for test and validation.*

We agree that if a large fraction of the input data was used for validation, then this procedure could certainly affect the fit. In the benchmarking analyses we conducted, we held out only 4% of the data. Importantly, in actual applications of scHPF, we do not actually use a validation set to assess performance, so this is not an issue.

- 11) *One way that HPF gets stuck in a local maximum, is by merging factors. In other words, it tends to find multiple near-identical factors, with near-identical beta and theta matrices. I suppose once two factors merge, they cannot diverge because locally they are already optimal. The result is in effect a reduction in the total number of factors, which may be fine of course. In my own code, I tried identifying redundant factors, and reinitializing them to a random state, but this did not help. Are redundant factors an indication that there is no more variance to be found?*

We have observed the issue to which you are referring. We find that it occurs in scRNA-seq datasets when, per the inference algorithm in the original HPF paper, phi is calculated

from the variational distributions for theta and beta during as the first step of the first iteration of training. As described in our expanded inference section, skipping this step during the first training iteration (or equivalently performing it as the last update) and instead initializing phi from a Dirichlet avoids “factor merging”. This may be because the randomness introduced when initializing theta and beta’s variational distributions is small compared to the randomness introduced by randomly initializing phi.

Revisions to text:

For scRNA-seq data, we have found that ϕ_{ig} ’s update order (relative to the other variational parameters) can affect symmetry breaking. In particular, performing (3) as the first step of the first iteration (before (1) and (2)) can result in redundant factors with similar weights across cells and genes.

- 12) *The hyperparameters are a bit confusingly named. For example, c is used to index the cells, and is also a hyperparameter, distinct from c' . I realize this is how it was done in the HPF paper, so maybe not a good idea to change now, but it would be helpful if you point out that c has double meaning.*

We agree and have revised the manuscript to eliminate this problem.

- 13) *Interestingly, your TensorFlow version is ~25% slower than my pure-python (numpy) version, even though TensorFlow uses all four of my cores, whereas numpy uses only one. It may be worth investigating if e.g. the datatypes matter (float64 vs float32 or even float16), if there's excessive copying of arrays somewhere, or something else.*

We agree that our TensorFlow implementation of scHPF is slow. We have rewritten our code and produced a new version using Numba that is dramatically faster. We have added a detailed assessment of computational performance to the manuscript (**Table 1**).

Revisions to text:

Because scHPF and ZINB-WaVE performed comparably in terms of predictive performance on the PBMC dataset, we carefully examined their respective factorizations in terms of computational expense and biological interpretability. For a single initialization with $K = 10$, training ZINB-WaVE took 7.38 hours and had a peak memory consumption of 17.8 Gb (**Table 1**). Using two threads reduced ZINB-WaVE’s runtime to just under four hours, but nearly doubled its memory consumption to 31.5 Gb. In contrast, our scHPF implementation took 2.5-10.7 minutes, depending on the number of threads available, and ~1.6 Gb of memory (**Table 1**). scHPF’s superior performance is in part due to optimized compilation and automatic parallelization with the Python Numba library (Lam et al, 2015). In addition, unlike ZINB-WaVE, scHPF only needs to consider nonzero matrix entries during training (Methods), which imparts a considerable theoretical advantage over methods that must iterate through (and in some cases store) every matrix entry.

Thank you for sending us your revised manuscript. We have now heard back from the two referees who were asked to evaluate your study. As you will see below, they think that the study has significantly improved as a result of the performed revision. They raise however a few remaining issues, which we would ask you to address in a minor revision.

Reviewer #1 is concerned about the validity of the survival analysis, which seems to be based on comparing patients with extreme expression levels of the signature. We would ask you to clarify how this analysis was performed and amend the related conclusions accordingly.

REFeree REPORTS

Reviewer #1:

The authors have done a thorough job in addressing my concerns with the original manuscript. The method proposed is novel, is competitive with status quo approaches in terms of analysis quality, and significantly outperforms status quo approaches in terms of time and memory costs (which is very important for scaling to emerging scRNA-seq "big data"). The use of stereotactic biopsies of human gliomas is commendable, and those novel data will provide an important resource. The manuscript, code and novel data will be of broad interest to the readership of MSB, as scRNA-seq is being widely adopted and their approach scales well and is therefore useful for meta-analysis.

My only lingering concern is the survival analysis. The approach taken by the authors is atypical. Most analyses of this type would divide the population into two cohorts (typically above and below median expression). When the authors tried that approach their signature was no longer significant. To achieve significance the authors need to use a threshold of at least 1.4. But, this results in 153/177 patients being excluded from consideration. Therefore their signature only correlates with survival when comparing the most extreme cases and disregarding the vast majority of the population. Even if this approach is considered valid, how can the authors be sure that there is sufficient statistical power in a comparison of 10 vs. 11 cases? I've never seen it done this way and I find the conclusion of the signature being prognostic to be somewhat misleading.

I would ask that the authors remove the survival analysis from the manuscript and abstract. I think the manuscript is strong enough on its own without it. If the authors feel strongly that the survival analysis should be retained, then I would ask that this portion of the manuscript specifically be reviewed by an epidemiologist to assess the correctness of this approach.

Minor: Maybe I couldn't find it, but the authors should include a statement of informed consent and IRB approval for the use of patient-derived tissues.

Reviewer #2:

The authors have adequately addressed all my questions and I have only one remaining concern:

In Fig. EV4, a couple of the factor designations are questionable, or not well supported:

The "pericytic" factor is probably more accurately described as "fibroblast" or perhaps "VLMC", given those collagens and Decorin. They likely also have Lum, Col1a2 and Pdgfra which are common in fibroblasts and VLMCs.

The "OPC-like" factor is marked by Olig1 (which is normally expressed in the whole oligodendrocyte lineage, not just OPCs) and Dll3 (which is specific to OPCs but also some neurons). More canonical markers would be Pdgfra and Cspg4, if those are indeed present in the factor. If they are not, I would doubt the designation as OPC-like.

Additional correspondence with the authors

11 January 2019

I would propose writing a response to the comment of reviewer #1 on the Kaplan-Meier plots and explaining in better detail how the analysis was done. If the way the analysis was performed can be clarified and if there are convincing arguments that the conclusions are justified, then I would see no reason to remove it. That being said, I do not think that removing it from the manuscript would be a major issue.

We noticed that the patients are split into "enriched", "depleted" and "intermediate" in terms of the expression of the signature, but the intermediate group is not shown at all in the Figure 4 plots. I would like to ask you if you could send us an edited figure in which all three groups are shown. I think that it would be useful to see this group in comparison to the other two (e.g. does it fall between the "enriched" and "depleted" groups?).

REVIEWER 1

My only lingering concern is the survival analysis. The approach taken by the authors is atypical. Most analyses of this type would divide the population into two cohorts (typically above and below median expression). When the authors tried that approach their signature was no longer significant. To achieve significance the authors need to use a threshold of at least 1.4. But, this results in 153/177 patients being excluded from consideration. Therefore their signature only correlates with survival when comparing the most extreme cases and disregarding the vast majority of the population. Even if this approach is considered valid, how can the authors be sure that there is sufficient statistical power in a comparison of 10 vs. 11 cases? I've never seen it done this way and I find the conclusion of the signature being prognostic to be somewhat misleading.

We disagree with the reviewer's assessment of our survival analysis on multiple points:

- 1) The approach that we took to divide TCGA cohort into groups is common, at least for gene expression data. For example, the cBIO Portal, a widely used web-based application for analyzing gene expression and clinical outcome from TCGA data (Cerami et al., 2012), also uses a z-score threshold for gene expression to divide patients into groups (Gao et al., 2013). This same approach of identifying sub-sets of patients based on an average expression threshold has been used in multiple studies to correlate gene expression with patient survival across the glioma cohort from TCGA (Flavahan et al., 2016; Patel et al., 2014).
- 2) There is a good rationale for not using the median expression level of a gene or gene signature to divide patient cohorts into phenotypically distinct groups. As mentioned in the previous rebuttal, it is easy to conceive of scenarios where the median-level approach would be problematic. For example, consider the case that the phenotype is overexpression of a gene-of-interest, and that this phenotype is relatively rare. Specifically, consider that its distribution across patients is $X = (0,0,1,1,1,1,1,1,1,1,2,2,2,1000,1000)$, where the patients with 1000 counts are of particular interest. The median of X is 1, and so the two groups would be $X_1 = (0,0,1,1,1,1,1,1,1,1)$ and $X_2 = (2,2,2,1000,1000)$, creating a situation where most of the subjects in the "high expression" group, X_2 , do not have the phenotype-of-interest (namely high expression of X).
- 3) We disagree that our "signature only correlated with survival when comparing the most extreme cases and disregarding the vast majority of the population". In Figure 4G, we showed a comparison between the two most extreme groups – the 10 patients with enrichment of our signature and the 11 patients with depletion. However, in Figure 4H, we also showed a comparison between the 10 patients with enrichment and all 151 remaining patients. In this analysis, where the results of our test are also highly significant, we do not disregard ANY patient in the cohort. All 161 patients are included in the analysis in Figure 4H. We have now added the population of patients that are neither enriched nor depleted in the gene signature to Figure 4G, which have a median survival that falls between those of the enriched and depleted patients and is significantly different from the enriched patients.

Minor: Maybe I couldn't find it, but the authors should include a statement of informed consent and IRB approval for the use of patient-derived tissues.

We included this statement in the "Author Checklist" which was included with our submission. We will add this statement to the Methods section as requested. We will add the following statement indicating IRB approval for the use of these specimens:

"Tissue was procured from de-identified patients who provided written informed consent to participate in these studies through a protocol approved by the Columbia Institutional Review Board (IRB-AAAJ6163)."

We do not have consent to share the WGS data for this patient under the IRB protocol and will state this in the Methods section as well under the subheading "Whole-genome sequencing".

REVIEWER 2

In Fig. EV4, a couple of the factor designations are questionable, or not well supported:

The "pericytic" factor is probably more accurately described as "fibroblast" or perhaps "VLMC", given those collagens and Decorin. They likely also have Lum, Coll1a2 and Pdgfra which are common in fibroblasts and VLMCs.

The "OPC-like" factor is marked by Olig1 (which is normally expressed in the whole oligodendrocyte lineage, not just OPCs) and Dll3 (which is specific to OPCs but also some neurons). More canonical markers would be Pdgfra and Cspg4, if those are indeed present in the factor. If they are not, I would doubt the designation as OPC-like.

We agree with the Reviewer that additional markers are needed to further support the designation of these cell types in Fig. EV4. To address, we have created Appendix Table S2 which has a longer list of genes associated with each factor. This list shows that additional genes such as *OLIG2* and *PDGFRA* are associated with the OPC-like factor, supporting this assignment. Furthermore, we also associate *PDGFRB*, which is a marker of pericytes with the factor assigned as "pericytes".

Cerami, E., Gao, J., Dogrusoz, U., Gross, B.E., Sumer, S.O., Aksoy, B.A., Jacobsen, A., Byrne, C.J., Heuer, M.L., Larsson, E., *et al.* (2012). The cBio cancer genomics portal: an open platform for exploring multidimensional cancer genomics data. *Cancer discovery* 2, 401-404.

Flavahan, W.A., Drier, Y., Liao, B.B., Gillespie, S.M., Venteicher, A.S., Stemmer-Rachamimov, A.O., Suva, M.L., and Bernstein, B.E. (2016). Insulator dysfunction and oncogene activation in IDH mutant gliomas. *Nature* 529, 110-114.

Gao, J., Aksoy, B.A., Dogrusoz, U., Dresdner, G., Gross, B., Sumer, S.O., Sun, Y., Jacobsen, A., Sinha, R., Larsson, E., *et al.* (2013). Integrative analysis of complex cancer genomics and clinical profiles using the cBioPortal. *Science signaling* 6, pl1.

Patel, A.P., Tirosh, I., Trombetta, J.J., Shalek, A.K., Gillespie, S.M., Wakimoto, H., Cahill, D.P., Nahed, B.V., Curry, W.T., Martuza, R.L., *et al.* (2014). Single-cell RNA-seq highlights intratumoral heterogeneity in primary glioblastoma. *Science (New York, NY)* 344, 1396-1401.

Corresponding Author Name: Peter A. Sims; David M. Blei
Journal Submitted to: Molecular Systems Biology
Manuscript Number: MSB-18-8857